# Boosting Domain Generalization in Object Detection through the Lens of Phase Invariance

## Abstract

Temporal and seasonal variations in dynamic real-world environments result in diverse visual appearances, posing significant challenges for object detection models to maintain consistently high performance. Although existing Domain Generalization (DG) methods have shown promise in enhancing model robustness, they often neglect the spatial structural relationships of objects during the learning of domain-invariant features, thereby limiting their effectiveness in object detection tasks compared to classification tasks. From the perspective of Preserving Phase Invariance (PPI), we propose a novel methodology that aims to enhance model generalization while preserving accurate object localization. This methodology comprises three complementary modules: Mix Normalization Perturbation (MNP), which synthesizes diverse styles to improve robustness; Sensitive Channel Perturbation (SCP), which suppresses domain-specific features at the channel level; and Amplitude-aware Attention (AOA), which applies spectral attention to the amplitude component. Together, these modules promote phase-invariant representations and contribute to improved cross-domain detection performance. Our approach fundamentally reduces the domain generalization gap in classification and detection by maintaining the integrity of key structural information. Our proposed methods achieve state-of-the-art performance on Unsupervised Domain Adaptation and Single Domain Generalization Object Detection benchmarks, even outperforming most recent state-of-the-art Domain Adaptation techniques. The code is available in the supplementary material.

## 1 Introduction

Domain generalization (DG) has progressed rapidly in image classification, where models learn to remain robust on unseen domains without access to target data (Guo et al., 2023b; Zhu et al., 2022b). It is natural to expect similar benefits for object detection, which is of greater practical importance. If classification can be made domain invariant, detection should improve as well. In practice, as shown in Figure 1, this expectation does not hold. When classification-oriented DG methods are applied to detection, gains are modest and can be negative. Under domain shift, we observe a consistent pattern. Class prediction inside ground-truth boxes improves, but the predicted boxes drift. Localization errors increase, and misaligned boxes offset the benefit from better recognition.

These observations motivate a more precise objective. Existing DG methods primarily enforce **image-level semantic consistency**. Object detection, by contrast, also requires **spatial structural consistency**. Classification asks *what*, while detection must also resolve *where*. Methods tailored to classification do not address this additional requirement, which explains their limited transfer to detection (Zheng et al., 2020; Li et al., 2022a; Liu et al., 2024). To close this gap, structural consistency should be enforced during feature extraction (Wu & Deng, 2022). The frequency-domain view via the Fourier transform makes the requirement concrete. Amplitude captures domain-specific style variation, while phase preserves spatial structure. Our design adopts **phase invariance (PPI)** as a principle. We keep phase stable across domains and allow amplitude to vary. The result is a representation that preserves geometry under shift while modeling style, which improves cross-domain detection.

In single-source DG detection, training must expose the model to enough appearance diversity to avoid source overfitting and must also steer the representation toward domain-invariant features that

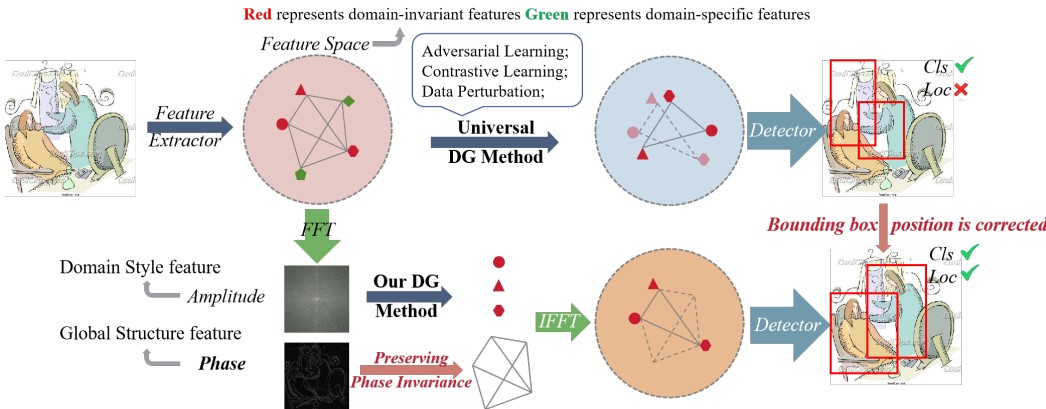

Figure 1: **The core idea of our method.** Conventional domain generalization (DG) methods primarily learn domain-invariant features but ignore spatial structure, leading to correct classification (Cls✓) but wrong localization (Loc✗) under domain shift. Our method leverages the fact that phase preserves spatial structure and enforces **phase invariance**. This guarantees structural consistency, thereby improving both classification and localization (Cls✓, Loc✓) in unseen domains.

stabilize localization. We formalize this as a trajectory from divergence to convergence. Under PPI, we implement this trajectory with three coordinated modules that respectively diversify features while preserving phase, perturb domain-sensitive channels to suppress spurious cues, and re-weight amplitudes in the frequency domain toward invariant bands.

First, the **mix normalization perturbation (MNP)** module induces controlled feature divergence at shallow layers to simulate latent domains. It perturbs feature statistics at the channel level and at the global level, synthesizing diverse and realistic styles (Nuriel et al., 2021; Yucel et al., 2023). Because these perturbations preserve phase, domain diversity increases without breaking spatial structure. Next, to enforce convergence toward domain-invariant representations in deeper layers, we introduce two complementary modules. The **sensitive channel perturbation (SCP)** module keeps a memory bank to track channels that are highly responsive to distributional change (Wu et al., 2021b;a). It then perturbs these domain-specific channels in the amplitude space, which shifts the detector toward stable channels that carry invariant information. In parallel, the **amplitude-aware attention (AOA)** module reweights the amplitude spectrum in the frequency domain to emphasize bands that encode domain-invariant cues, reinforcing convergence to structurally consistent features. Related adversarial approaches exist (Fan et al., 2021; Yang et al., 2021; Zhu et al., 2022a), but they do not explicitly enforce phase invariance.

In this paper, we study single-source domain generalization for object detection and introduce a phase-invariant, frequency-domain approach. Our contributions are $i$) a diagnosis of why classification-oriented DG transfers poorly to detection by showing that enforcing image-level semantic consistency is not sufficient and that detection also requires spatial structural consistency, $ii$) a principle we call Preserving Phase Invariance that constrains learning to keep phase fixed while allowing amplitude to vary so geometry is preserved while style can change, and $iii$) a divergence-to-convergence framework that implements this principle with Mix Normalization Perturbation for feature diversification and Sensitive Channel Perturbation plus Amplitude-aware Attention for convergence, supported by extensive experiments that show improved generalization on multi-weather and real-to-art benchmarks, including a $26\%$ mAP improvement across five target domains on SDGOD and strong results across one stage, two stage, and transformer based detectors.

## 2 MOTIVATION FOR PPI IN OBJECT DETECTION

### 2.1 WHY CLASSIFICATION-ORIENTED DG OFTEN FAILS FOR DETECTION

A straightforward strategy for improving detector robustness is to borrow techniques that work well for classification (e.g., style augmentation, adversarial Learning, invariant risk minimization). Intuitively this transfer should succeed: if a model can ignore domain styles and keep semantics, detection should benefit as well. Empirically, however, this expectation breaks down. Using the TIDE error decomposition toolbox to analyze a range of state-of-the-art DG methods applied to detection, we

observe a consistent pattern: these methods typically *increase* the per-object classification accuracy while concurrently *degrading localization quality*, and the net effect is often a drop in overall detection performance (see Table. 2.2. A lower value indicates fewer errors of that type, meaning better performance, and vice versa.). This pattern indicates a fundamental mismatch between the objectives of classification DG and the needs of detection.

## 2.2 PHASE INVARIANCE PRESERVES STRUCTURAL CONSISTENCY

To maintain structural consistency, we naturally turn to a basic property in the frequency domain. An image $I(x, y)$ can be represented by its Fourier transform

$$F(u, v) = A(u, v) e^{i\phi(u,v)},$$

where $A(u, v)$ denotes the amplitude spectrum and $\phi(u, v)$ the phase spectrum. While $A(u, v)$ primarily encodes domain-specific styles such as color and texture, the phase $\phi(u, v)$ determines the spatial alignment of edges and contours, and hence object structure. Preserving phase invariance thus provides a principled and

| Method | Main Errors | | | | mAP50 |
| | Cls | Loc | Both | Miss | |
|---|---|---|---|---|---|
| Baseline | 7.31 | 8.12 | 1.18 | 21.74 | 44.6 |
| DomainDrop | 6.11 | 12.74 | 1.20 | 21.12 | 43.2 |
| **After PPI** | **6.01** | **6.33** | **1.14** | **21.11** | **46.8** |
| AdvST | 5.87 | 12.09 | 1.68 | 23.65 | 43.4 |
| **After PPI** | **6.27** | **6.29** | **1.39** | **22.28** | **46.4** |

Table 1: TIDE analysis on the C2F benchmark. We compare classification-based DG methods on detection performance and show their improvements when constrained by our proposed PPI paradigm.

direct way to ensure structural consistency under domain shifts. We further validate this hypothesis empirically. As shown in Table. 2.2, when classification-based DG methods that previously underperform in detection are constrained by PPI, their localization accuracy improves markedly and overall detection performance increases significantly. This confirms that PPI is both theoretically sound and broadly effective as a general constraint for DG methods.

## 2.3 TOWARD A DIVERGENCE–CONVERGENCE FRAMEWORK FOR SDGOD

Building on the PPI paradigm, we further consider the single-source DG detection (SDGOD) setting, where the challenge is that images from a single source domain lack the features of the object in different domains, making it difficult for the model to determine domain-invariant features from them. This motivates a framework that guides domain features from *divergence to*

| Method | Main Errors | | | | mAP50 |
| | Cls | Loc | Both | Miss | |
|---|---|---|---|---|---|
| C-GAP | 12.6 | 14.4 | 1.88 | 24.74 | 35.5 |
| UFR | 10.12 | 13.82 | 1.87 | 23.12 | 38.2 |
| DivAlign | 10.34 | 13.09 | 1.68 | 23.42 | 37.5 |
| **PPI-3(OURS)** | **6.87** | **6.11** | **0.92** | **22.68** | **43.4** |

Table 2: DWD results with TIDE analysis of performance differences between our method and SOTA SDGOD methods.

*convergence*, as illustrated in Table. 2.3. Finally, to highlight the superiority of our approach, we compare it with other state-of-the-art SDG detection methods using TIDE analysis. Our method achieves better classification accuracy, higher localization precision, and consistently higher mAP scores across domains. These results reinforce the necessity of PPI and set the stage for the framework details described in the following section.

## 3 PROPOSED METHOD

From the perspective of Preserving Phase Invariance (PPI), we construct a detection framework consisting of three modules, designed to significantly improve the performance on the SDGOD task, which we refer to as **PPI-3**. We presents an overview of the proposed framework in Algorithm 1.

---
**Algorithm 1** Forward pass with PPI-3
---
**Require:** image $x$, modules `Stem`, `Stages`, `MNP`, `SCP`, `AOA`, `Detector`
**Ensure:** detections $y$
1: **compute** initial features $z \leftarrow \texttt{Stem}(x)$
2: **update** $z \leftarrow \texttt{MNP}(z)$
3: **update** $z \leftarrow \texttt{Stage}(z)$
4: **update** $z \leftarrow \texttt{SCP}(z)$
5: **update** $z \leftarrow \texttt{AOA}(z)$
6: **compute** $y \leftarrow \texttt{Detector}(z)$
7: **return** $y$
---

## 3.1 MIX NORMALIZATION PERTURBATION

Our method first introduces a feature divergence module, Mix Normalization Perturbation (MNP), which generates richer domain style features and thereby encourages the model to learn domain-invariant representations. MNP is composed of INP and LNP, which alter image styles, and an Occlusion Perturbation (OP) module that simulates partial image occlusions. In Appendix C, we theoretically analyze how the MNP module can directly modify the distribution of the amplitude spectrum, enabling style transfer across domains without altering phase information. In this subsection, we focus on the design and implementation of the corresponding modules. Transferring the original domain style to other styles using normalization typically requires statistics from the target domain to guide the transfer direction. However, this is not applicable in SDG tasks. We therefore employ perturbation values sampled from a specific noise distribution as random transfer directions. To ensure that the transferred domain features cover more potential styles, we proposed methods to perturb feature map statistics along different dimensions: INP, which changes the statistics of channels, and LNP, which alters the global statistics of the feature map. Images generated by blending these two types of perturbations not only exhibit color variations but also more authentically reflect changes in overall pixel values—indicative of varying illumination conditions—and incorporate blur effects attributable to light scattering.

Figure 2: MNP module: generating diverse styles through mixed perturbations.

We consider a feature map $x \in \mathbb{R}^{C \times H \times W}$. The per-channel (instance) mean and standard deviation over spatial indices are $\mu_i, \sigma_i \in \mathbb{R}^C$, and the layer-wide mean and standard deviation over all entries are $\mu_l, \sigma_l \in \mathbb{R}$. Let $w_1, w_2 \in [0, 1]$ with $w_1 + w_2 = 1$, and draw random coefficients $\alpha_1, \beta_1 \in \mathbb{R}^C$ and $\alpha_2, \beta_2 \in \mathbb{R}$ (unit-mean). Define $\hat{\mu}_i = \alpha_1 \mu_i$, $\hat{\sigma}_i = \beta_1 \sigma_i$, $\hat{\mu}_l = \alpha_2 \mu_l$, $\hat{\sigma}_l = \beta_2 \sigma_l$. The feature divergence reads

$$y_{\text{style}} = w_1 \Big( \hat{\sigma}_i \, \frac{x - \mu_i}{\sigma_i} + \hat{\mu}_i \Big) + w_2 \Big( \hat{\sigma}_l \, \frac{x - \mu_l}{\sigma_l} + \hat{\mu}_l \Big),$$

where multiplications with $\mu_i, \sigma_i$ act channelwise and with $\mu_l, \sigma_l$ act as scalars.

We apply **MNP** to the shallow layers of CNNs, which benefits from the nonlinear capabilities of CNNs to generate a wide array of potential styles within the high-dimensional feature space. For better visual understanding, we use feature inversion techniques (Zeiler & Fergus, 2014) to translate the perturbed feature maps back into the input image space. As shown in Figure 7, styles generated through our approach are not only rich and diverse but also more accurately reflect real-world appearances, all while preserving clear instance contour structures.

Although the styles synthesized through this method cover a wide array of scenarios, we observed that in some extreme cases, such as the presence of raindrops, the structure in partial regions of an object can be entirely disrupted. To bolster the generalization capabilities of our detection algorithm, spectively; and $\leq, <$ are comparison operators defining the block boundaries.

We apply an occlusion perturbation (OP) that replaces entries covered by randomly sampled rectangular blocks with a fixed value while leaving all other entries unchanged. Given an input feature map $F$ and an occlusion value $v_{\text{occ}}$, draw $N_{\text{blocks}} \sim \mathcal{U}_{\text{int}}[N_{\min}, N_{\max}]$ and, for each block $k$, sample a top–left corner $(y_{s,k}, x_{s,k})$ and size $(h_k, w_k)$. The output $F'$ is

$$F'_{b,c,y,x} = F_{b,c,y,x} + (v_{\text{occ}} - F_{b,c,y,x}) \, \mathbb{I} \Bigg( \bigvee_{k=1}^{N_{\text{blocks}}} \{ y_{s,k} \leq y < y_{s,k} + h_k \ \wedge \ x_{s,k} \leq x < x_{s,k} + w_k \} \Bigg),$$

where $\mathbb{I}(\cdot)$ is the indicator (1 if the condition holds, 0 otherwise), $\vee$ and $\wedge$ denote logical OR and AND, and the inequalities define half-open ranges that specify each block's interior.

## 3.2 SENSITIVE CHANNEL PERTURBATION

MNP's diverse features facilitate learning of domain-invariant features. However, we observed that MNP's varied stylized outputs render some deep-layer neurons highly sensitive (evidencing large activation variance), contrasting with more stable ones. We hypothesize sensitive channels capture domain-specific information, while stable channels learn domain-invariant features. To steer the network towards these stable, domain-invariant features, our proposed SCP module further perturbs the sensitive channels, diminishing the network's reliance on their content for inference. Crucially, to preserve structural information within these sensitive channels, this perturbation employs randomized normalization, adhering to the PPI paradigm.

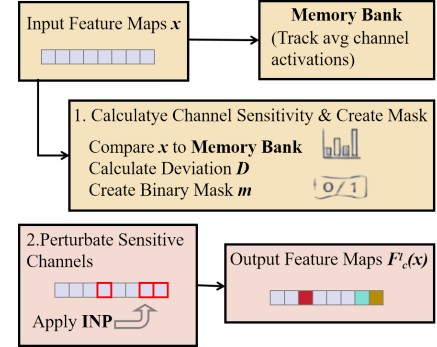

Figure 3: SCP module: perturbing domain-sensitive channels guided by a memory bank.

As depicted in Figure 3, this approach involves decomposing the feature maps into individual channels, selecting those sensitive to domain variations for perturbations, and then merging all channels. The procedure is as follows: Initially, we set up a memory bank to record the activation levels of each channel for unperturbed source domain images.

For each stage $l$ and channel $c$ we maintain a memory-bank mean $\text{MB}_c^l$ of activations. Given sample $x_i$ with activation $F_c^l(x_i)$, update the bank by an exponential moving average with coefficient $\alpha \in [0, 1]$,

$$\overline{\text{MB}}_c^l = \alpha \, \text{MB}_c^l + (1 - \alpha) \, F_c^l(x_i),$$

and form the deviation

$$D_c^l(x_i) = \left| \overline{\text{MB}}_c^l - F_c^l(x_i) \right|.$$

Let $\tau_j^l(x_i)$ be the $(100-j)$th percentile of $\{D_k^l(x_i)\}_{k=1}^C$ (so at least $j\%$ of channels satisfy $D_k^l(x_i) \geq \tau_j^l(x_i)$). Define the binary mask

$$m_c^l(x_i) = \begin{cases} 0, & D_c^l(x_i) \geq \tau_j^l(x_i), \\ 1, & \text{otherwise}, \end{cases}$$

so $m_c^l(x_i) = 0$ characterize the top $j$ percentage most domain-sensitive channels and $m_c^l(x_i) = 1$ the rest.

While DomainDrop (Guo et al., 2023a) is a known technique that eliminates channels capturing domain-specific features, we argue that removing channels entirely can result in the loss of important structural data, adversely affecting the model's detection capabilities. Moreover, selecting channels based on the difference in activation values lacks interpretability, making it difficult to guarantee that channels chosen in each training phase indeed contain domain-specific features.

We perturb only the channels flagged as domain-sensitive by the mask ($m_c^l(x_i) = 0$) and leave the others unchanged. For sample $x_i$ at stage $l$, define

$$\widehat{F}_c^l(x_i) = m_c^l(x_i) \, F_c^l(x_i) + \left(1 - m_c^l(x_i)\right) \text{INP}\!\left(F_c^l(x_i)\right), \qquad c = 1, \ldots, C,$$

where $\text{INP}(\cdot)$ is the normalized perturbation operator introduced above (applied with broadcasting over spatial indices). Thus channels with $m_c^l(x_i) = 1$ pass through unchanged, while those with $m_c^l(x_i) = 0$ are replaced by their perturbed versions.

### 3.3 AMPLITUDE-AWARE ATTENTION

While SCP indirectly prompts the network's deeper layers to rely on domain-invariant features for inference, we also desire the model to learn domain-invariant features more directly. Therefore, we consider introducing an attention mechanism, where the attention map acts as a cue to guide the model's focus towards regions with a higher concentration of these invariant features. However, directly incorporating an attention module into the feature extractor can lead to significant information loss and impair the model's generalization capability. To address this, we are the first to propose an attention mechanism implemented on the amplitude spectrum within the frequency domain. This

mechanism is designed to capture domain-invariant features from different frequency bands across various channels, without sacrificing the object's structural features. Figure 4 illustrates the AOA mechanism. Initially, the features are projected into an embedding space using a 1x1 convolutional layer for further processing.

Given a spatial feature map $f \in \mathbb{R}^{M \times N}$, let $F(u, v)$ be its 2D Fourier transform and define $A(u, v) = |F(u, v)|$ and $P(u, v) = \text{atan2}\big(\Im F(u, v), \Re F(u, v)\big)$, where $\Re$ and $\Im$ denote real and imaginary parts. Let $r = \sqrt{u^2 + v^2}$ and $R = \frac{1}{2}\sqrt{M^2 + N^2}$. Segment the amplitude radially into three bands:

$$A_{\text{seg}}(u, v) := \begin{cases} A_{\text{low}}(u, v) & = A(u, v)\,\mathbb{I}(r < R/3), \\ A_{\text{mid}}(u, v) & = A(u, v)\,\mathbb{I}(R/3 \leq r < 2R/3), \\ A_{\text{high}}(u, v) & = A(u, v)\,\mathbb{I}(r \geq 2R/3), \end{cases} \tag{1}$$

with $\mathbb{I}(\cdot)$ the indicator function and $M, N$ the height and width of $f$.

For the attention mechanism, we use the spatial attention module from CBAM (Woo et al., 2018), chosen for its simplicity and effectiveness. It's important to note that the attention module's implementation is not the primary focus of our approach.

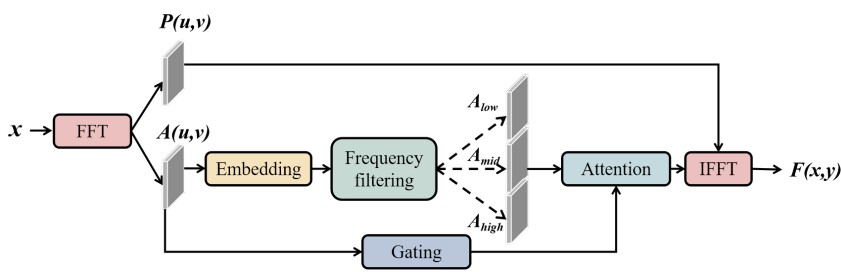

Figure 4: AOA module: emphasizing domain-invariant frequency bands in the amplitude spectrum.

We utilize the entire amplitude spectrum as a prior input to a gating MLP, which in turn modulates the attention strength applied to different frequency bands. We modulate the amplitude spectrum with attention weights and keep the phase unchanged, then reconstruct via an inverse FFT. Let the attention weights be $\text{CBAM}(A_{\text{seg}}(u, v))$ and let $\text{Gate}(\cdot)$ act pointwise on amplitudes. The reconstructed feature map is

$$F(x, y) = \text{IFFT}\Big(\text{Gate}\big(\text{CBAM}(A_{\text{seg}}(u, v))\, A(u, v)\big)\, e^{iP(u, v)}\Big).$$

By moving away from the traditional application of attention modules in the spatial domain, our approach emphasizes the amplitude spectrum. This focus is particularly advantageous for tasks aimed at domain generalization, offering a more effective strategy for detecting domain-general features.

## 4 RELATED WORK

### 4.1 DOMAIN GENERALIZATION

In recent years, there has been a significant increase in methods aimed at addressing Out-of-Distribution (OOD) challenges (Guo et al., 2023b; Lin et al., 2021; Mahajan et al., 2021), focusing on two primary strategies: Domain Generalization (DG) (Li et al., 2023; Guo et al., 2023a) and Domain Adaptation (DA) (Long et al., 2013; Glorot et al., 2011). DG aims to prepare Deep Neural Networks (DNNs) for generalization across various domains by training on one or multiple source domains without prior knowledge of the target domain. DG methods are broadly categorized into data augmentation (Volpi & Murino, 2019; Yucel et al., 2023) and domain-invariant feature learning (Fan et al., 2021; Yang et al., 2021; Sun et al., 2021). Data augmentation typically involves transferring styles across multiple domains at the image-level through techniques such as Variational Autoencoder (VAE) (Huang & Belongie, 2017) or Generative Adversarial Networks (GAN) (Zhu et al., 2017). A significant contribution by Fan et al. (Fan et al., 2023), enhanced style diversity at the feature level by applying feature perturbation methods to DNNs' shallow layers, aiming to increase data diversity and thereby improve model generalization. Meanwhile, domain-invariant feature learning, employs

methods such as feature disentanglement(Wu & Deng, 2022; Wu et al., 2021b;a), meta-learning (Dou et al., 2019; Zhang et al., 2022; Wei et al., 2021), and adversarial learning (Fan et al., 2021; Yang et al., 2021; Zhu et al., 2022a) to teach models to recognize and learn features common across different domains, enabling them to generalize to unseen domains effectively. *However, these methods, being inherently task-specific and predominantly tailored for classification tasks, frequently neglect the crucial aspect of preserving structural integrity and accurate feature localization. This oversight renders their straightforward application to object detection tasks impractical.*

## 4.2 SINGLE DOMAIN GENERALIZATION OBJECT DETECTION

Single-Domain Generalization (SDG) (Volpi et al., 2018) addresses the challenge of learning from only one annotated source domain and generalizing to unseen domains. Prior works have explored augmentation and regularization, such as adversarial training to diversify inputs (Qiao et al., 2020) and network regularization for feature alignment (Lee et al., 2023). While these methods have shown success in classification, progress in Single-Domain Generalization Object Detection (SDGOD) is still limited. Wu (Wu & Deng, 2022) proposed cyclic-disentangled self-distillation, Vidit (Vidit et al., 2023) employed visual-linguistic prompts to enrich features, and Liu (Liu et al., 2024) applied causal modeling to mitigate bias. Despite these advances, most approaches overlook the additional structural requirements of detection beyond classification.

Meanwhile, several studies have examined generalization from the frequency perspective. FFDI (Wang et al., 2022) leverages mid-frequency contours to enhance high-frequency details, UAV-OD (Wang et al., 2023) applies learnable filters on the amplitude spectrum to separate invariant from domain-specific features, HybridAugment++ (Yucel et al., 2023) reduces reliance on high-frequency components for robustness, and DFF introduces frequency-based attention to boost classification. These works demonstrate the promise of frequency modeling but primarily focus on amplitude manipulation. In contrast, our approach emphasizes the critical role of *phase invariance* in preserving structural consistency for detection—a perspective not previously explored. Building on this paradigm, we design a divergence-to-convergence framework that balances feature diversity and invariant representation learning, substantially improving generalization in SDGOD.

## 5 EXPERIMENTS

### 5.1 DATASETS AND PERFORMANCE METRICS

We evaluate our method on three benchmarks with significant domain gaps: Cityscapes → FoggyCityscapes (C2F) (Cordts et al., 2016; Sakaridis et al., 2018), the Diversity Weather dataset (Wu & Deng, 2022), and the Real-to-Artistic dataset (Inoue et al., 2018). The first two benchmarks focus on weather variations in real urban scenes, while the third tests generalization under extreme real-to-artistic shifts. Performance is measured by mAP@0.5 following standard protocols. Details of each dataset are provided in Appendix H.

### 5.2 IMPLEMENTATION DETAILS

Our main baseline is Faster R-CNN with a ResNet-50 backbone implemented in Detectron2 (Wu et al., 2019; He et al., 2016). We further verify our method on RetinaNet (one-stage) and DETR (transformer-based) to evaluate generality. All experiments follow standard DG protocols (e.g., NP (Fan et al., 2023)): training for a fixed number of epochs, averaging over multiple runs, and excluding target-domain validation data to avoid bias. Training hyperparameters and module-specific implementations are reported in Appendix D.

### 5.3 COMPARISON WITH STATE-OF-THE-ART

**C2F.** *Goal: test robustness under synthetic weather shift.* Table 3 shows that PPI-3 achieves 48.8 mAP, surpassing all DG and DA baselines, even though DA methods (e.g., DA-Faster (Chen et al., 2018), ICCR-VDD (Wu et al., 2021b), CIGAR (Liu et al., 2023), CSDA (Gao et al., 2023), PT (Chen et al., 2022)) exploit target-domain data. On the DG side, PPI-3 consistently outperforms strong methods such as IBN (Pan et al., 2018), SFA (Li et al., 2021), MixStyle (Zhou et al., 2021), DSU (Li et al., 2022b), and PØDA (Fahes et al., 2023). In particular, our MNP module alone already outperforms the latest style synthesis method NP+ (Fan et al., 2023) by +0.9 mAP. **Observation:**

Table 3: UDA/DG object detection AP50 performance on Faster-RCNN detector.

| Synthesis Method | C2F | | DG Method | C2F |
|---|---|---|---|---|
| Baseline | 22 | | BIN | 28.4 |
| Image SR | 30.4 | | IBN | 31.2 |
| NP/NP+ | 44/46.3 | | SFA | 25.3 |
| **DA Method** | | | pAdaIN | 27.6 |
| DA-Faster | 32 | | Mixstyle | 30.1 |
| ICCR-VDD | 40 | | DSU | 34.1 row |
| CIGAR | 44.9 | | PØDA | 47.4 |
| CSDA | 45.8 | | **PPI-3 (Ours)** | **48.8** |
| PT | 47.1 | | | |

Table 4: Single domain generalization results on Diversity Weather Dataset.

| Method | mAP | | | | | |
|---|---|---|---|---|---|---|
| | day clear | night clear | dusk rainy | night rainy | day foggy | mean |
| FR | 48.1 | 34.4 | 26 | 12.4 | 32 | 30.5 |
| SW *ICCV'19* | 50.6 | 33.4 | 26.3 | 13.7 | 30.8 | 30.9 |
| IBN-Net *ECCV'18* | 49.7 | 32.1 | 26.1 | 14.3 | 29.6 | 30.3 |
| IterNorm *CVPR'19* | 43.9 | 31.8 | 27.3 | 12.6 | 28.4 | 28.8 |
| ISW *CVPR'21* | 51.3 | 33.2 | 25.9 | 14.1 | 31.8 | 31.2 |
| S-DGOD *CVPR'22* | 56.1 | 36.6 | 28.2 | 16.6 | 33.5 | 34.2 |
| C-Gap *CVPR'23* | 51.3 | 36.9 | 38.1 | 24.1 | 37.2 | 35.5 |
| DivAlign *CVPR'24* | 52.8 | 42.5 | 32.3 | 18.7 | 38.5 | 37.5 |
| UFR *CVPR'24* | 58.6 | 40.8 | 33.2 | 19.2 | 39.6 | 38.2 |
| **PPI-3 (Ours)** | **62.0** | **45.4** | **43.5** | **24.5** | **41.7** | **43.4** |

These results confirm that phase-preserving perturbations are more effective than purely style-transfer methods. Interestingly, DA methods fail to outperform our approach despite using target supervision, suggesting that structure-preserving generalization is more crucial than explicit adaptation under foggy conditions.

**Diversity Weather.** *Goal: evaluate single-domain generalization under diverse conditions.* Table 4 shows that PPI-3 consistently outperforms strong SDGOD baselines such as UFR (Liu et al., 2024), DivAlign (Danish et al., 2024), C-Gap (Vidit et al., 2023), and Single-DGOD (Wu & Deng, 2022). On average, PPI-3 improves over UFR by +5.2 mAP across five unseen domains, with the largest gain (+10.3) under dusk rainy. **Observation:** The strongest gains occur in scenarios with complex weather mixtures (rain + low light), where style variations are large but structural cues remain essential. This indicates that PPI's preservation of structural consistency is especially valuable under compound shifts.

**Real-to-Artistic.** *Goal: test robustness under extreme real-to-artistic shifts.* Table 5 shows improvements of +10.8, +14.2, and +11.5 mAP on Clipart1k, Watercolor2k, and Comic2k respectively, substantially outperforming NP and other baselines. **Observation:** Unlike weather shifts, artistic shifts dramatically alter textures and color palettes, but preserve coarse structures. PPI-3 leverages this property, retaining structural consistency and yielding large improvements. This suggests that PPI is particularly advantageous when amplitude/style varies drastically but phase/structure is stable.

## 5.4 ABLATION STUDY

We conduct ablation studies with the objective of verifying three aspects: (i) the necessity of all modules, (ii) the rationality of our design choices, and (iii) the architecture-agnostic property of our method.

**Module necessity.** Table 7 reports incremental results on Diversity Weather. MNP alone improves generalization, confirming the role of divergence. Adding AOA yields large gains in dusk rainy ($40.1 \rightarrow 44.0$) but smaller gains in night clear ($43.5 \rightarrow 43.8$), suggesting frequency-based attention is more effective under style-rich shifts. SCP further stabilizes performance by suppressing unstable channels; Fig. 8(b) shows reduced channel variance, and Fig. 8(c) reveals that AOA mainly attends to low-frequency regions. **Observation:** These patterns support the divergence–convergence principle: MNP broadens exposure, AOA strengthens stable frequencies, and SCP mitigates channel-level instability.

**Design choices.** We further analyze the impact of critical design decisions in PPI-3, focusing on noise sampling, channel suppression, and attention location.

*Noise sampling.* Table 8 compares different noise distributions within MNP. We find that performance is relatively stable across distributions, but $G(1, 0.75)$ gives the best results. More importantly, mixed normalization (combining LN and IN perturbations) consistently outperforms single normalization. **Observation:** Single normalization alters feature statistics in only one direction, limiting style diversity. By mixing multiple normalizations, we create richer and more realistic style variations without disturbing phase, which broadens the detector's exposure to domain shifts while preserving structural consistency. This confirms the role of MNP as an effective divergence mechanism.

Table 5: Performance comparison with baseline and possible ablations, mAP@0.5(%) reported. The model is trained on Pascal VOC and tested on Clipart1k, Watercolor2k and Comic2k.

| Method | VOC | Clipart | Watercolor | Comic |
|---|---|---|---|---|
| Faster R-CNN | 81.8 | 25.7 | 44.5 | 18.9 |
| NP | 78.4 | 27.1 | 52.4 | 19.8 |
| UFR | 78.0 | 33.0 | 57.1 | 25.9 |
| **PPI-3** | **77.5** | **36.5** | **58.7** | **30.3** |

Table 6: Performance of our method on different detection frameworks, including one-stage and transformer-based detector.

| Method | mAP | | | | | |
| | DC | NC | DR | NR | DF | mean |
|---|---|---|---|---|---|---|
| RetinaNet | 54.7 | 37.9 | 27.5 | 10.8 | 35.7 | 33.3 |
| **PPI-3** | **55.0** | **44.6** | **37.2** | **19.5** | **40.5** | **39.4** |
| Detr | 49.1 | 33.8 | 24.5 | 10.4 | 33.1 | 30.2 |
| **PPI-3** | **49.6** | **37.3** | **30.9** | **16.9** | **37.5** | **34.2** |

Table 7: Comparative Analysis of Channel Suppression Methods, Variations in Attention Module Positions, and Incremental Experiments across Different Approaches.

| Row | Method | mAP | | | | | |
| | | DC | NC | DR | NR | DF | mean |
|---|---|---|---|---|---|---|---|
| 1 | Baseline | 48.1 | 34.4 | 26.0 | 12.4 | 32.0 | 30.50 |
| 2 | **MNP** | 61.4 | 43.5 | 40.1 | 21.5 | 39.9 | 41.28 |
| 3 | MNP+AOF | 55.1 | 35.5 | 33.6 | 16.8 | 34.6 | 35.12 |
| 4 | MNP+AOP | 46.6 | 29.8 | 26.5 | 12.9 | 30.1 | 29.18 |
| 5 | **MNP+AOA** | 61.7 | 43.8 | **44.0** | 23.8 | 41.1 | 42.88 |
| 6 | MNP+AOA+SCP(DROPOUT) | 62.3 | 44.4 | 42.6 | 23.1 | 42.3 | 42.94 |
| 7 | MNP+AOA+SCP(MEAN) | 62.2 | 44.5 | 43.0 | 23.8 | **42.4** | 43.18 |
| 8 | **MNP+AOA+SCP(INP)** | **62.0** | **45.4** | 43.5 | **24.5** | 41.7 | **43.42** |

Table 8: Ablation studies on noise hyperparameters of our MNP method. IN Noise and LN Noise represent the distributions of noise in INP and LNP sampling, respectively. The last two columns represent the use of only one perturbation.

| IN Noise | U(0,2) | G(1,0.5) | G(1,0.75) | G(1,0.75) | G(1,0.75) | G(1,1) | - | G(1,0.75) |
|---|---|---|---|---|---|---|---|---|
| LN Noise | U(0,2) | G(1,0.5) | G(1,0.5) | G(1,0.75) | U(0,2) | U(0,2) | U(0,2) | - |
| C2F | 46.4 | 46.4 | 47.4 | 47.1 | 47.2 | 46.4 | 43.3 | 44.8 |

*Channel suppression.* We compare Dropout-based suppression with our INP strategy on the Diversity Weather dataset (Table 7). Dropout directly removes sensitive channels, yielding lower mAP (43.18, 42.94), whereas INP replaces their activations with stable averages, achieving higher mAP (43.42, 43.44). **Observation:** Removing entire channels discards useful structural cues and destabilizes learning, while INP reduces sensitivity without erasing information. This aligns with PPI: the goal is not to delete features but to reweight them in a way that preserves phase-related structural information while dampening domain-specific amplitude noise.

*Attention location.* We evaluate placing attention on the frequency spectrum (AOF), amplitude spectrum (AOA), and phase spectrum (AOP). As shown in Table 7, only AOA improves generalization, while AOF and AOP hurt performance. **Observation:** Applying attention on frequency or phase directly disturbs structural alignment, violating the PPI constraint. In contrast, amplitude attention selectively emphasizes invariant frequency bands, reinforcing stable structure without breaking phase. Visualization in Fig. 8(c) shows that AOA focuses on low-frequency regions, which contain robust domain-invariant cues. This analysis supports our choice of amplitude-only attention as the convergence step.

**Architecture generality.** Table 6 shows PPI-3 consistently improves Faster R-CNN, RetinaNet, and DETR. Interestingly, the relative improvement is largest for DETR, suggesting transformer-based detectors, which rely on global attention, benefit most from explicit structural constraints. **Observation:** PPI-3 is architecture-agnostic and complements modern attention-based models particularly well.

## 6 CONCLUSION

In this paper, we explore model generalization with a focus on maintaining the integrity of instance structural information. We observe that while most domain generalization (DG) methods excel in classification tasks, they often fall short in detection tasks due to a lack of consideration for instance structure. To tackle this challenge, we propose a new framework aimed at Preserving Phase Invariance (PPI) to safeguard structural information. This framework consists of three key modules: domain synthesis with Mixed Normalization Perturbation (MNP), Sensitive Channel Perturbation (SCP), and Amplitude-aware Attention (AOA), all designed to improve the domain generalization capabilities of detection models. Our experimental results show that this approach outperforms existing domain adaptation and generalization methods in various contexts. In the future, we plan to further investigate PPI-based DGOD methods and encourage the research community to apply more sophisticated DG strategies to detection tasks leveraging the PPI concept.

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

## A  APPENDIX

## B  TECHNICAL APPENDICES AND SUPPLEMENTARY MATERIAL

In the Appendix, we have provided detailed supplements for the omitted sections in the main text. In Section C, we have utilized the formula derivation of features in the frequency domain to prove that Normalization does not alter the target structure. In Section D, we have elaborated on the implementation of our method and the settings of experimental hyperparameters. In Section E, We present a comparison of the mean Average Precision achieved by our method for each object category against State-Of-The-Art (SOTA) methods on the Diversity Weather Dataset. In Section F, We add more experiments on the value selection of $j$ in SCP module. In Section G, we explored which original error cases were improved after using the PPI constraint method. In Section H, We provide a detailed description of the three datasets that are mentioned in the main text and used in our experiments. In Section I, we have presented visualizations of real urban scene images under different weather conditions along with the detection results of PPI-3. In Section J, we discuss the computational performance and real-time performance of our approach. Finally in section K, we discussed some limitations that our current method still faces in cross-domain scenarios, while also highlighting its potential contributions to future development.

## C  PROOF OF NORMALIZATION PRESERVING TARGET STRUCTURE

To demonstrate how perturbing images through normalization alters their statistical properties while preserving the target structure, let's consider the original image denoted as $I(x, y)$, which undergoes a Fourier transformation expressed as:

$$\mathcal{F}(u,v) = \sum_{x=0}^{H-1} \sum_{y=0}^{W-1} I(x,y) exp^{-j2\pi\left(\frac{x}{H}u + \frac{y}{W}v\right)} \tag{2}$$

Here, $u$ and $u$ denote the horizontal and vertical frequency coordinates, respectively, and $W$ and $H$ represent the image's width and height. Then we perform image transformations involving scaling and translation defined as:

$$\hat{I}(x,y) = aI(x,y) + b, \quad \hat{\mu} = a\mu + b, \quad \hat{\sigma}^2 = a^2\sigma^2 \tag{3}$$

Where a is the scaling factor, and b is the translation amount. Performing a Fourier transform on the new $\hat{I}(x,y)$ yields:

$$\begin{aligned}
\hat{\mathcal{F}}(u,v) &= \iint [aI(x,y) + b] exp^{-j2\pi(ux+vy)} \, \mathrm{d}x\mathrm{d}y \\
&= a \iint I(x,y) exp^{-j2\pi(ux+vy)} \, \mathrm{d}x\mathrm{d}y + b \iint exp^{-j2\pi(ux+vy)} \\
&= a\,\mathcal{F}(u,v) + b\delta(u,v)
\end{aligned} \tag{4}$$

Here, $\delta(u,v)$ represents the Dirac delta function, which is a constant. The phase is solely determined by the ratio component of the complex number. Therefore, the original phase remains consistent with the phase of the transformed image. The new amplitude can be obtained by computing the modulus.

$$\hat{\mathcal{A}}(u) = |a\mathcal{F}(u,v) + b\delta(u,v)| \tag{5}$$

It is evident that the scaling factor $a$ and $\delta(u,v)$ affect the magnitude's value. In the main paper, we have explained that the amplitude component directly reflects the image style, while the phase component reflects the target structure. Therefore, we have demonstrated through the aforementioned formula that perturbing image statistics through normalization can achieve style transfer without altering the structural content of the image.

## D  MORE DETAILS FOR THE METHOD

Our method PPI-3 is designed as a divergent-to-convergent process. In our framework, our modules are only applied to the backbone of the detection model. The input features are randomly migrated

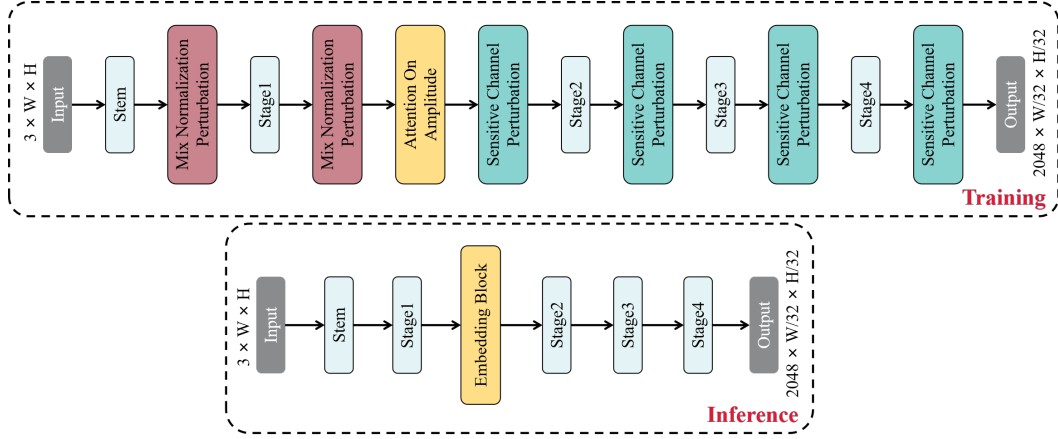

Figure 5: The flowchart of the PPI-3 backbone in our method consists of two sections: the training process at the top and the inference process at the bottom. It integrates ResNet-50 with three modules that we proposed, each functioning at different locations.

to features of different domain styles by the Mix Normalization Perturbation (MNP) module in the shallow layers. Then, in the subsequent deep layers, the Sensitive Channel Perturbation (SCP) and Attention On Amplitude (AOA) modules are applied to extract domain-invariant features as the main features for inference by the detection head. As shown in Figure 5, we use ResNet-50 as the backbone to explain the respective locations where our three designed modules function. ResNet-50 He et al. (2016) mainly consists of a stem block and four stage blocks. During the training phase, we apply MNP after the first stem and the first stage. AOA is applied after the first stage, while SCP is applied after each stage. The main reason for adding only one AOA module is that we do not want the model to overly favor low-frequency components (as explained in the main text, the AOA module focuses on low-frequency components). During the inference stage, we only retain the Embedding block in the AOA module to ensure that our method barely increases the computational burden of the original detection model. Additionally, researchers can easily reproduce our modules through the schematics and formulas for each method provided in the main text or refer to our released code links. It is worth noting that in the MNP module, we set the mixing ratio of features obtained from two different regularization perturbations as $w1 = w2 = 0.5$. This is because both perturbations have strong randomness, and this is a parameter that cannot be learned during training.

To ensure the stability of training, the Memory Bank in the SCP module need acquire more accurate channel activation values at the outset. Therefore, when adding the SCP module, we divide the training into two stages. In the first stage, we train with the MNP and AOA modules for a set number of epochs, and then add the SCP module to continue training for a specific number of additional epochs. Table 9 and Table 10 present the settings of our experimental hyperparameters on two benchmark datasets. The only difference lies in the number of training epochs, as the Diversity Weather Dataset has more training data, thus requiring more epochs to fully learn the data characteristics.

## E    MORE DETAILS FOR THE EXPERIMENT

To better analyze the generalization ability of PPI-3 to unseen domains, we discuss the per-class results of the individual target domains on the Diversity Weather Dataset.

### E.1    DAYTIME CLEAR TO DAY FOGGY.

Compared to other scenarios, objects in foggy images appear more evenly blurred, with significant loss of contours and textures. As shown in Table 12, our method achieves an average improvement in detection performance across most categories, except for a slight decrease compared to C-Gap on the Bike category.

| Config | MNP or MNP+AOA | PPI-3(MNP+A0A+SCP) |
|---|---|---|
| description | Default object detection setting | |
| codebase | Detectron2-CycConf Confusion | |
| pretraining data | ImageNet | |
| backbone | ResNet-50 | |
| backbone norm | FrozenBN | |
| backbone freeze_at | 2 | |
| detector | Faster R-CNN with FPN | |
| batch size | 16 | |
| # GPU | 2 | |
| LR scheduler | WarmupMultiStepLR | |
| base LR | 0.02 | |
| gamma | 0.1 | |
| momentum | 0.9 | |
| weight decay | 0.0001 | |
| warmup method | linear | |
| warmup iters | 1000 | |
| warmup factor | 1.0 / 1000 | |
| **LR steps** | (12000, 16000) | - |
| **training iterations** | 17000 | 4500 |
| training (min, max) size | (800, 1333) | |
| testing (min, max) size | (800, 1333) | |

Table 9: Cityscapes Cordts et al. (2016) to FoggyCityscapes Sakaridis et al. (2018) experiment setting details.

| Config | MNP or MNP+AOA | PPI-3(MNP+A0A+SCP) |
|---|---|---|
| description | Default object detection setting | |
| codebase | Detectron2-CycConf Confusion | |
| pretraining data | ImageNet | |
| backbone | ResNet-50 | |
| backbone norm | FrozenBN | |
| backbone freeze_at | 2 | |
| detector | Faster R-CNN with FPN | |
| batch size | 16 | |
| # GPU | 2 | |
| LR scheduler | WarmupMultiStepLR | |
| base LR | 0.02 | |
| gamma | 0.1 | |
| momentum | 0.9 | |
| weight decay | 0.0001 | |
| warmup method | linear | |
| warmup iters | 1000 | |
| warmup factor | 1.0 / 1000 | |
| **LR steps** | (24000, 32000) | - |
| **training iterations** | 37500 | 17500 |
| training (min, max) size | (800, 1333) | |
| testing (min, max) size | (800, 1333) | |

Table 10: Diversity Weather Dataset Vidit et al. (2023) experiment setting details.

### E.2 DAYTIME CLEAR TO NIGHT CLEAR.

In the Night clear scenario, images exhibit significant loss of contours and darkened colors. The black background and the contours of the targets form a challenging situation for detection tasks. However, as shown in Table 13, our method outperforms other current methods in all categories, especially in the Car and Person categories, where there is a substantial improvement in metrics.

### E.3 DAYTIME CLEAR TO DUSK RAINY.

Raindrops are a type of high-frequency noise that disrupts the appearance and structure of objects by altering the refraction of light. However, in our method, the AOA module effectively addresses

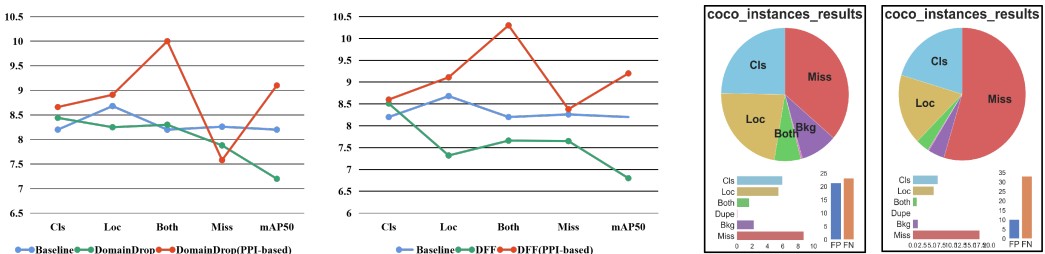

(a) Performance analysis on cross-domain task          (b) Detection error analysis by Tide toolbox

Figure 6: (a) Comprehensive analysis of detection algorithm performance under domain shift. Using the TIDE toolbox Bolya et al. (2020), we compute error types including localization, classification, both, and missed detections, along with mAP, all normalized to the same scale. (b) Raw analysis results of TIDE toolbox on DomainDrop.

this type of high-frequency noise. Consequently, as shown in Table 14, our method demonstrates significant improvements in every category in the Dusk Rainy scenario.

### E.4  DAYTIME CLEAR TO NIGHT RAINY.

Night Rainy is the most challenging scenario among the four target domains. It is not only affected by dim lighting but also accompanied by uneven raindrops that disrupt the appearance of objects in the image. Comparing the metrics with other scenarios, it is evident that detectors often struggle to achieve good detection results in this scenario. Nevertheless, as shown in Table 15, our method has achieved significant improvements in most categories except for Motor. Compared to the previously best method, C-Gap, our method PPI-3 have improved by 30% (24.3 vs. 18.7)."

## F  ABLATION STUDY ON SCP

In the SCP module, we need to set the ratio of suppressed channels. In theory, we need to limit the perturbed channels to the channels of the learning domain-related features as much as possible without affecting the channels of the learning domain-invariant features. Therefore, we conducted an ablation experiment on the parameter $j$, as shown in the following figure.

As expected, when the value of $j$ increases, the generalization ability of the model will improve. However, when the value of $j$ is too large, it will affect the normal learning of the model and make it impossible to continue training.

| $j$ | =0 | =0.1 | =0.2 | =0.3 | >=0.4 |
|---|---|---|---|---|---|
| **mAP** | 42.3 | 42.7 | 43.1 | 43.4 | NAN |

Table 11: The impact of different $j$ values on performance

## G  ANALYZE THE REASONS FOR PERFORMANCE IMPROVEMENT

As shown in Figure 6(b), We use Tide toolbox to analyze the difference in detecting major errors between methods with and without PPI constraints. We compare our SCP module in PPI-3 with the DomainDrop method on the "night rainy" domain in the Diversity Weather Dataset. For fair comparison, our two methods are MNP+SCP and MNP+DomainDrop. The training parameters are consistent with the parameters of the model trained on the Diversity Weather Dataset benchmark in the paper. The mAP50 scores are 24.3 for PPI versus 22.0 for DomainDrop. Regarding the main errors, the classification dAP is 6.4 (PPI) versus 6.0 (DomainDrop), and the localization dAP is 5.2 (PPI) versus 6.0 (DomainDrop). This indicates that our method PPI, while slightly increasing the classification error, has improved over all detection performance by locating more objects.

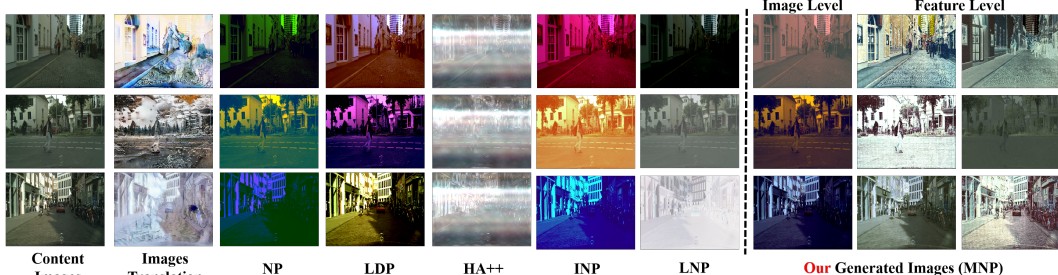

Figure 7: **Visualization of different style synthesis methods.** The second column shows the results obtained by training CUT Park et al. (2020) with winter and watercolor styles as the target domains. The third to fifth columns show the style transfer effects brought to images by several different SOTA feature perturbation methods Fan et al. (2023); Li et al. (2023); Yucel et al. (2023). The sixth to eighth columns illustrate the effects of applying INP alone, LNP alone, and their combination. The ninth and tenth columns demonstrate the style transfer effects of MNP's perturbations at the feature level.

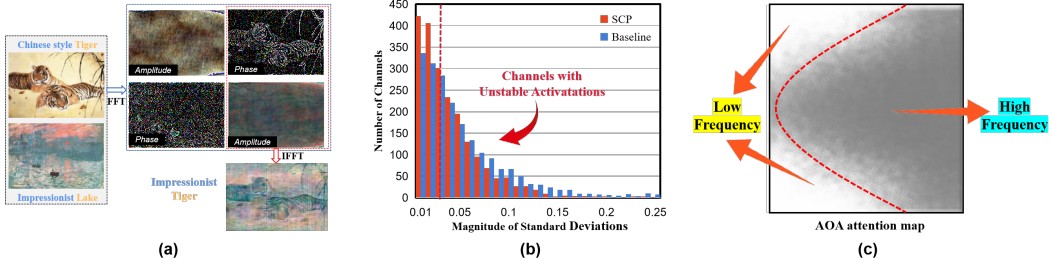

Figure 8: (a) Synthesizing a new Image by merging frequency domain components: the process combines the phase spectrum of a Chinese-style tiger with the amplitude spectrum of an Impressionist-style lake, using FFT and IFFT to create an 'Impressionist Tiger' (b) The robustness of channels to domain shifts before and after **SCP** is applied. (c) Attention maps learned by the attention modules in AOA. White represents 1, while black represents 0.

## H    MORE DETAILS FOR THE DATASET

**For the C2F experiment,** Cityscapes dataset, which includes 2,975 training and 500 validation images from urban settings across 50 cities in standard weather conditions, was used. FoggyCityscapes, synthesized by adding artificial fog to the Cityscapes images, maintaining the same training-validation split.

**The Diversity Weather Dataset** contains images under five distinct weathers: clear during the day, clear at night, rainy at dusk, rainy at night, and foggy during the day. It integrates data from Cityscapes, Berkeley Deep Drive 100K (BBD-100K) Yu et al. (2020), and Adverse-Weather datasets Hassaballah et al. (2020). Our model was trained on 19,395 images depicting daytime clear conditions and evaluated across all five weather scenarios, which include 8,313 daytime clear, 26,158 night clear, 3,501 dusk rainy, 2,494 night rainy, and 3,775 daytime foggy images.

**The Real to Artistic Dataset** consists of four datasets, including the Pascal VOC which contains real images from 20 different object classes and acts as a source domain, and remaining three are Clipart1k, Watercolor2k and Comic2k which offer artistic and comic images and are used as target domains. Clipart1k shares the same 20 classes as Pascal VOC, while Watercolor2k and Comic2k consist of 6 classes each, which are subsets of the Pascal VOC classes.

In these experiments, we followed the experimental protocols established by prior studies Fan et al. (2023); Wu & Deng (2022); Vidit et al. (2023) and evaluated the models using Average Precision at 50% intersection over union (AP50) for each category.

## I VISUALIZATION OF DETECTION RESULTS

To better illustrate the generalization ability of our model and provide a more intuitive understanding of real-world single-source domain generalization for object detection, we present images from the training and testing scenarios of both the C2F and Diversity Weather Dataset experiments in Figure 9 and Figure 10, respectively. Additionally, the detection performance of our method on the target domains is shown in the last block. In both domain generalization experiments, our training scenarios are clear daytime images, which are commonly used for detection tasks. However, due to the diversity of the real world, weather conditions constantly change, and it is difficult to predict and capture the characteristics of unknown weather. As demonstrated by the target scenarios in the experiments, target objects become difficult to discern in these scenarios due to significant changes in contours and textures. This poses a challenging but necessary task for detection models. Our proposed PPI-3 method significantly enhances the generalization ability of the detection model. From the visualized detection results, it can be seen that our method can detect most target objects in each scenario.

## J COMPUTATIONAL EFFICIENCY AND REAL-TIME APPLICABILITY

The efficiency of our framework is a notable advantage. Among the three modules in PPI-3, only the Attention on Amplitude module adds 66,402 extra parameters, which is only a 0.2% increase compared to the baseline (Faster R-CNN). The other modules only involve matrix operations, resulting in an additional computational load of 19 GFLOPs. Training time increases by 15%, while inference time only increases by 5%. Compared to the methods we mentioned in main paper, our method show clear advantages; for example, C-Gap(CVPR'2023) adds 33,172,979 parameters—an increase of 80%. A detailed comparison table will be included in the revised version. Therefore, in the deployment of real-time detection algorithms, the inference time of our framework is entirely dependent on the original inference time of the improved baseline. For example, on the RTX 3090, Fast-RCNN runs at approximately 40 FPS, while PPI-3 based on Fast-RCNN runs at approximately 38 FPS.

## K LIMITATION

Integrating spatial attention modules into the amplitude spectrum has improved the model's ability to generalize across different scenarios, particularly against high-frequency noise common in environmental disturbances. However, this approach has been less effective against low-frequency noise, such as lighting variations. We initially hypothesized that attention modules could dynamically adjust their focus across frequency bands to mitigate a range of interferences. Yet, the vast diversity in these interferences presents a significant challenge to achieving this adaptability through existing spatial attention mechanisms alone. Our proposed AOA points towards a novel research direction, aiming to refine how attention mechanisms are employed on the amplitude spectrum for domain generalization, thereby seeking more effective strategies to cover a broader spectrum of environmental interferences.

## L THE USE OF LARGE LANGUAGE MODELS

We lightly used LLM to help us polish the sentences we had already completed to make the article more readable.

| Method | AP | | | | | | | mAP |
|---|---|---|---|---|---|---|---|---|
| | Bus | Bike | Car | Motor | Person | Rider | Truck | ALL |
| Faster R-CNN | 30.7 | 26.7 | 49.7 | 26.2 | 30.9 | 35.5 | 23.2 | 31.9 |
| SW Pan et al. (2019) | 30.6 | 26.2 | 44.6 | 25.1 | 30.7 | 34.6 | 23.6 | 30.8 |
| IBN-Net Pan et al. (2018) | 29.9 | 26.1 | 44.5 | 24.4 | 26.2 | 33.5 | 22.4 | 29.6 |
| IterNorm Huang et al. (2019) | 29.7 | 21.8 | 42.4 | 24.4 | 26 | 33.3 | 21.6 | 28.4 |
| ISW Choi et al. (2021) | 29.5 | 26.4 | 49.2 | 27.9 | 30.7 | 34.8 | 24 | 31.8 |
| S-DGOD Wu & Deng (2022) | 32.9 | 28 | 48.8 | 29.8 | 32.5 | 38.2 | 24.1 | 33.5 |
| C-Gap Vidit et al. (2023) | 36.1 | **34.3** | 58 | 33.1 | 39 | 43.9 | 25.1 | 38.5 |
| **PPI-3(Ours)** | **38.5** | 33.7 | **63.9** | **37.7** | **43.8** | **44.8** | **28** | **41.5** |

Table 12: **Per-class results on Daytime Clear to Day Foggy.**

| Method | AP | | | | | | | mAP |
|---|---|---|---|---|---|---|---|---|
| | Bus | Bike | Car | Motor | Person | Rider | Truck | ALL |
| Faster R-CNN | 37.7 | 30.6 | 49.5 | 15.4 | 31.5 | 28.6 | 40.8 | 33.5 |
| SW | 38.7 | 29.2 | 49.8 | 16.6 | 31.5 | 28 | 40.2 | 33.4 |
| IBN-Net | 37.8 | 27.3 | 49.6 | 15.1 | 29.2 | 27.1 | 38.9 | 32.1 |
| IterNorm | 38.5 | 23.5 | 38.9 | 15.8 | 26.6 | 25.9 | 38.1 | 29.6 |
| ISW | 38.5 | 28.5 | 49.6 | 15.4 | 31.9 | 27.5 | 41.3 | 33.2 |
| S-DGOD | 40.6 | 35.1 | 50.7 | 19.7 | 34.7 | 32.1 | 43.4 | 36.6 |
| C-Gap | 37.7 | 34.3 | 58 | 19.2 | 37.6 | 28.5 | 42.9 | 36.9 |
| **PPI-3(Ours)** | **44.8** | **39.8** | **71.4** | **19.7** | **55.0** | **35.2** | **49.3** | **45.0** |

Table 13: **Per-class results on Daytime Clear to Night Clear.**

| Method | AP | | | | | | | mAP |
|---|---|---|---|---|---|---|---|---|
| | Bus | Bike | Car | Motor | Person | Rider | Truck | ALL |
| Faster R-CNN | 36.8 | 15.8 | 50.1 | 12.8 | 18.9 | 12.4 | 39.5 | 26.6 |
| SW | 35.2 | 16.7 | 50.1 | 10.4 | 20.1 | 13 | 38.8 | 26.3 |
| IBN-Net | 37 | 14.8 | 50.3 | 11.4 | 17.3 | 13.3 | 38.4 | 26.1 |
| IterNorm | 32.9 | 14.1 | 38.9 | 11 | 15.5 | 11.6 | 35.7 | 22.8 |
| ISW | 34.7 | 16 | 50 | 11.1 | 17.8 | 12.6 | 38.8 | 25.9 |
| S-DGOD | 37.1 | 19.6 | 50.9 | 13.4 | 19.7 | 16.3 | 40.7 | 28.2 |
| C-Gap | 37.8 | 22.8 | 60.7 | 16.8 | 26.8 | 18.7 | 42.4 | 32.3 |
| **PPI-3(Ours)** | **47.5** | **33.5** | **77.2** | **20.9** | **46.7** | **27.7** | **53.2** | **43.8** |

Table 14: **Per-class results on Daytime Clear to Dusk Rainy.**

| Method | AP | | | | | | | mAP |
|---|---|---|---|---|---|---|---|---|
| | Bus | Bike | Car | Motor | Person | Rider | Truck | ALL |
| Faster R-CNN | 22.6 | 11.5 | 27.7 | 0.4 | 10 | 10.5 | 19 | 14.5 |
| SW | 22.3 | 7.8 | 27.6 | 0.2 | 10.3 | 10 | 17.7 | 13.7 |
| IBN-Net | 24.6 | 10 | 28.4 | 0.9 | 8.3 | 9.8 | 18.1 | 14.3 |
| IterNorm | 21.4 | 6.7 | 22 | 0.9 | 9.1 | 10.6 | 17.6 | 12.6 |
| ISW | 22.5 | 11.4 | 26.9 | 0.4 | 9.9 | 9.8 | 17.5 | 14.1 |
| S-DGOD | 24.4 | 11.6 | 29.5 | **9.8** | 10.5 | 11.4 | 19.2 | 16.6 |
| C-Gap | 28.6 | 12.1 | 36.1 | 9.2 | 12.3 | 9.6 | 22.9 | 18.7 |
| **PPI-3(Ours)** | **33.4** | **13.7** | **51.1** | 4.9 | **25.1** | **11.5** | **30.1** | **24.3** |

Table 15: **Per-class results on Daytime Clear to Night Rainy.**

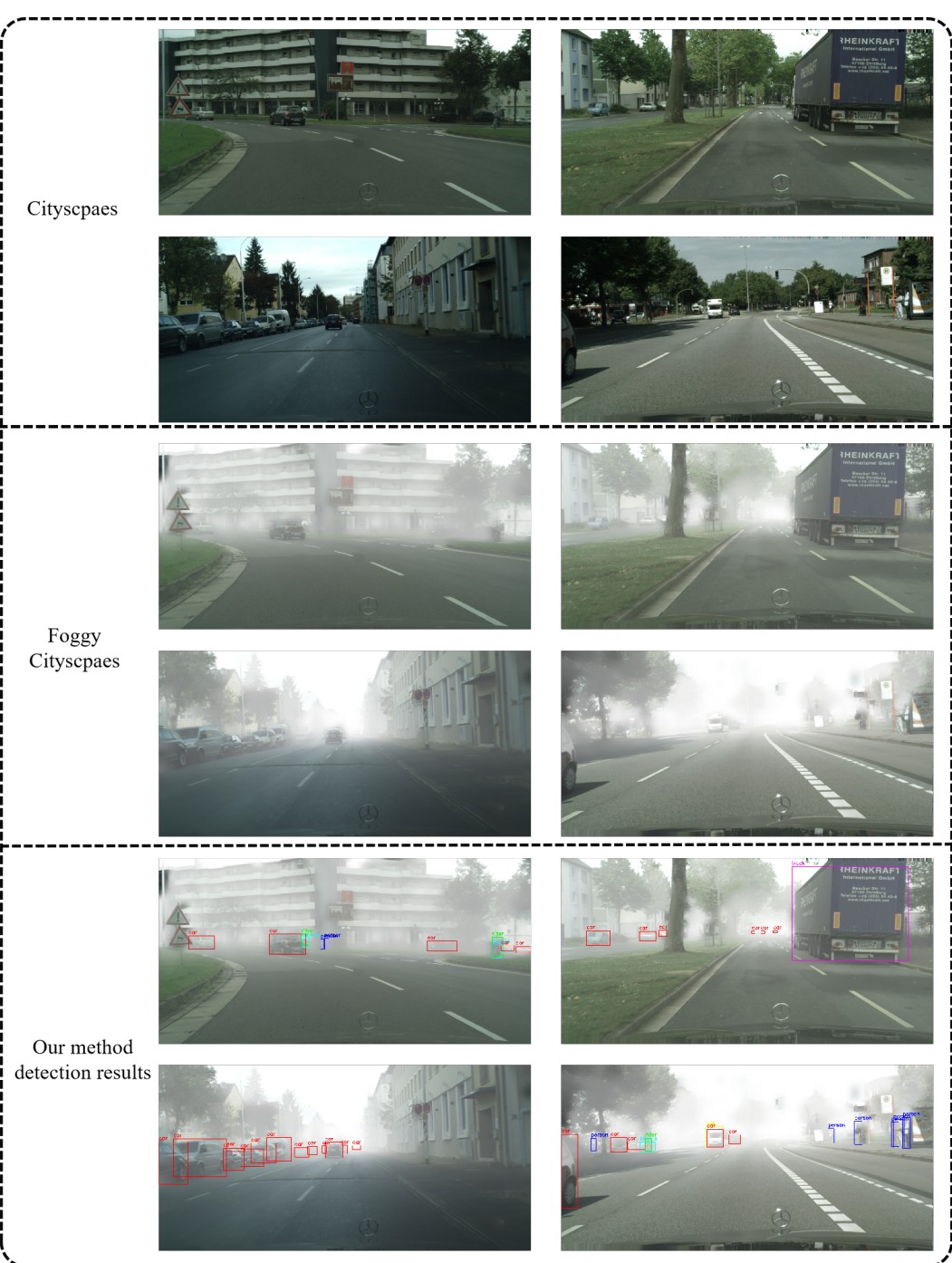

Figure 9: Images from Cityscapes and FoggyCityscapes scenarios, where the model is trained in clear weather conditions and adapted to foggy weather conditions. The last block demonstrates the detection performance of our method on the four target domains.

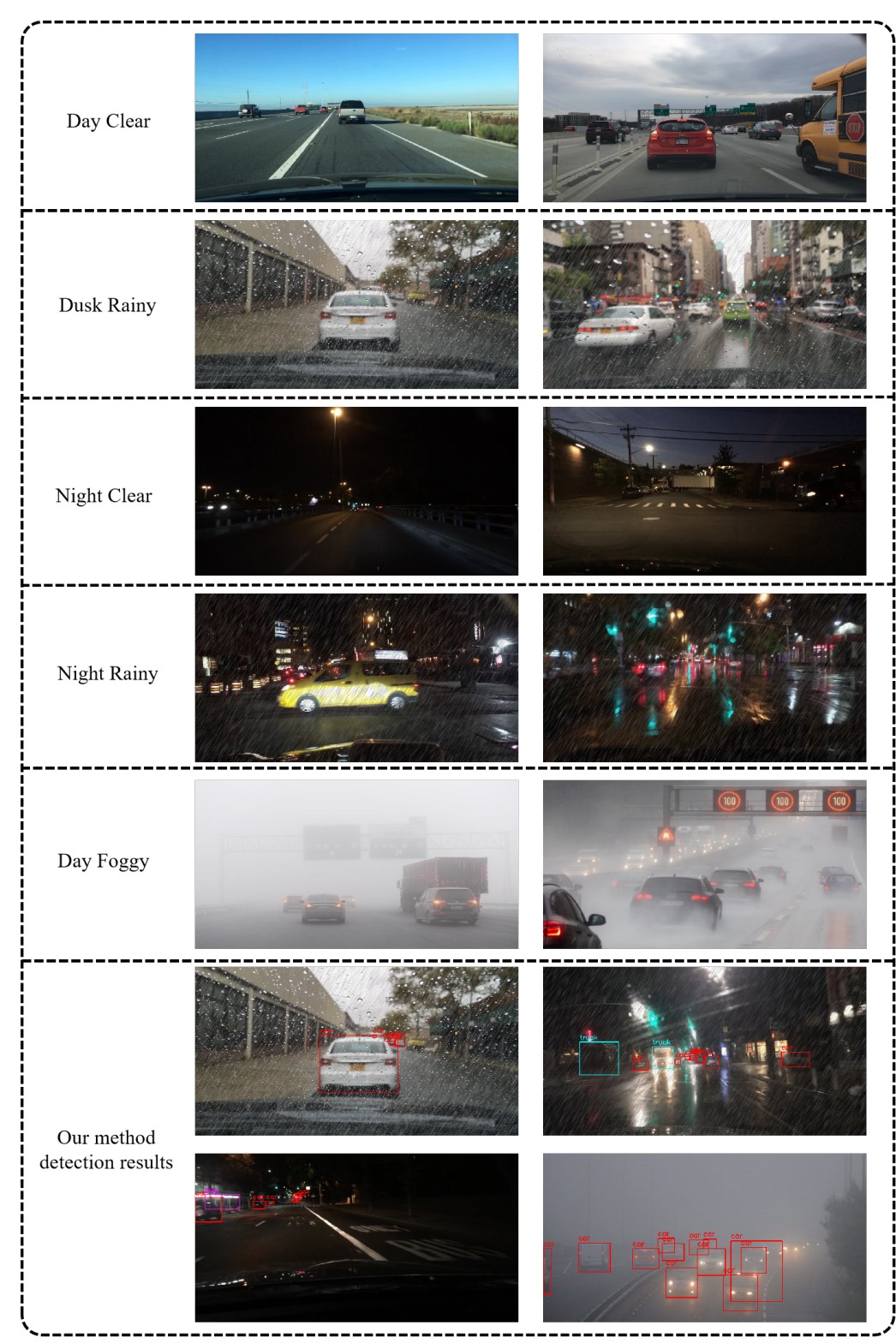

Figure 10: The images in the Diversity Weather Dataset cover five weather scenarios, including the training domain 'Day Clear' and four target domains: 'Dusk Rainy', 'Night Clear', 'Night Rainy', and 'Day Foggy'. The last block presents the detection performance of our method on the four target domains.

