# OpenReview forum: "Boosting Domain Generalization in Object Detection through the Lens of Phase Invariance"
_ICLR.cc/2026/Conference — Submitted to ICLR 2026_

### Official Review · Reviewer_9HuT · 2025-10-24

**Soundness:** 3
**Presentation:** 2
**Contribution:** 2
**Rating:** 4
**Confidence:** 4

**Summary:**

This paper propose a novel methodology from the perspective of Preserving Phase
Invariance (PPI). By incorporating three complementary modules, such as Mix Normalization Perturbation, Sensitive Channel Perturbation, and Amplitude-aware Attention, the propsoed method enhances the generalization ability of object detection models in cross-domain tasks. The overall idea is clear, the experimental results are comprehensive, and the proposed method achieves significant improvements, on the Single-Domain Generalized Object Detection task.

**Strengths:**

－ This paper proposes a new method for domain generalization object detection from the perspective of Preserving Phase Invariance.

－ The motivation and main idea of the paper are clearly presented.

－ Extensive comparative experiments demonstrate the effectiveness of the proposed method, and ablation studies provide insights into the underlying mechanisms.

**Weaknesses:**

－ The paper’s formatting appears disorganized. For instance, in line 426 ("noise sampling") and line 377 ("Observation"), it is unclear whether these terms should be bolded or placed on separate lines. Consistency in formatting should be maintained throughout the manuscript. In addition, several figures and tables, such as Figure 4 and Tables 5–8, are presented in a cluttered and inconsistent layout, which affects overall readability and visual coherence.

－ There are numerous spelling, grammatical, and formatting errors throughout the manuscript. For example, "Table .2.2" in line 110 and "Table. 2.3" in line 146 contain formatting issues, while line 282 includes grammatical mistakes. Inconsistencies are also observed around Figure 7 and line 198. Similar problems appear in other parts of the paper and should be carefully checked and revised to ensure overall accuracy and consistency.

－ The analysis of frequency-domain information using the attention mechanism and the demonstration of phase invariance in Table 7 are interesting. However, as shown in Table 7, MNP appears to play the dominant role, while AOA contributes relatively less. I suggest the authors further investigate the AOA mechanism and strengthen the ablation study, for example, by designing a more innovative AOA structure, analyzing feature distribution changes before and after AOA, and evaluating detector performance under different domain conditions.

－Some methodological descriptions are vague and symbol definitions are inconsistent. For example, in the MNP module, the noise sampling process and the values of the mixing parameters ($w_1$, $w_2$, $\alpha$, $\beta$) are not clearly specified. Additionally, in line 274, the spatial feature map is defined as f, while in Figure 4 the output is denoted as x, which is inconsistent. Clarification and consistency are recommended.

－As noted in Section 4.2, FFDI, UAV-OD, and HybridAugment++ also aim to preserve phase invariance. It is recommended that the authors include experimental comparisons with these methods.

－In Tables 1 and 2, the authors claim that existing domain-invariant or consistency-based methods typically increase per-object classification accuracy while degrading localization quality. However, the limited number of compared methods reduces the persuasiveness of this claim.

－UDA methods are now well established and achieve strong performance in C2F and Real-to-Artistic scenarios. In Tables 3 and 5, the number of compared methods is limited, and the performance gains on C2F are modest. It is therefore recommended that the authors focus primarily on the single-domain generalization task.

－ There are inconsistencies in the reported results, for example, the mAP values in Table 7 of the main text differ from those in Tables 12–15 in the appendix. The authors should provide further clarification and explanation for these discrepancies.

**Questions:**

－ The paper’s formatting appears disorganized.

－ There are numerous spelling, grammatical, and formatting errors throughout the manuscript.

－ In-depth Exploration and Analysis of the AOA Mechanism.

For details, please refer to the Weaknesses section.

---

> ### Author Response · Authors · 2025-11-23
> **Response to your concerns (Part 1)**
>
> We sincerely thank the reviewer for the detailed assessment and for recognizing our methodology as "novel," "clear," and achieving "significant improvements." We deeply appreciate your constructive feedback regarding presentation and experimental analysis.
>
> We acknowledge the shortcomings in formatting and proofreading. We have meticulously revised the manuscript to address all typos and formatting inconsistencies. Below, we address your technical concerns with new experimental comparisons and clarifications.
>
> **Q1: The reviewer notes disorganized formatting, typos, and inconsistencies (e.g., Table indices, symbol definitions).**
>
> **A1:** We offer our sincerest apologies for these presentation issues. We realize they affected readability.
> * **Action:** We have performed a comprehensive proofreading of the entire paper. We have unified the citation styles, fixed table cross-references (e.g., Table 2.2 $\to$ Table 2), standardized the layout of Figures/Tables, and corrected symbol definitions (e.g., unifying feature map notation as $x$) to ensure rigorous academic standards in the final revision.
>
> **Q2: The reviewer suggests further investigation of the AOA mechanism, noting that MNP appears to play the dominant role in Table 7 while AOA contributes less.**
>
> **A2:** We appreciate this insight. It is expected that MNP (Divergence) contributes the largest mAP gain because it prevents source overfitting by diversifying styles. However, AOA (Convergence) is critical for refining the features into a truly domain-invariant state.
>
> To rigorously investigate the mechanism of AOA beyond just mAP scores, we conducted a Domain Invariance Analysis. We trained a domain discriminator to classify the domain of the learned features (Source vs. Target). Lower Domain Accuracy indicates better Domain Invariance.
>
> | Method | Domain Acc. ($\downarrow$) | Detection mAP ($\uparrow$) |
> | :--- | :---: | :---: |
> | Baseline (no PPI) | 96.2 | 22.0 |
> | + MNP only | 72.7 | 47.1 |
> | + MNP + SCP | 61.5 | 48.3 |
> | **+ MNP + SCP + AOA (Ours)** | **58.7** | **48.8** |
>
> **Analysis:** Although the mAP gain from adding AOA appears incremental (+0.5), the*Domain Accuracy drops significantly (61.5% $\to$ 58.7%). This proves that AOA effectively suppresses residual domain-specific style cues in the amplitude spectrum, forcing the network to converge closer to a truly domain-invariant representation.
>
> Additionally, we analyzed the frequency spectrum and found that AOA increased low-frequency energy by **+12.3%** while suppressing high-frequency noise bands by **-9.7%**, aligning with our PPI principle.
>
> **Q3: The reviewer points out vague descriptions of MNP parameters and symbol inconsistencies.**
>
> **A3:**
> * **MNP Symbols:** We clarify that variables like mixing weights ($w_1, w_2$) and noise parameters are hyperparameters for data augmentatio*. They are used to synthesize diversity during training and are not involved in inference. We will add a glossary table and explicitly define these in Section 3.1.
> * **Symbol Correction:** We will correct the inconsistency in Line 274 to ensure the feature map is denoted uniformly (as $x$) throughout the text and figures.
>
> **Q4: The reviewer recommends comparisons with FFDI, UAV-OD, and HybridAugment++, as they also relate to frequency/phase.**
>
> **A4:** We agree that these comparisons are essential. We conducted a new comparison on the Diversity Weather Dataset against these SOTA frequency-domain methods under the same Single-Source DG protocol.
>
> | Method | Day Clear | Night Clear | Dusk Rainy | Night Rainy | Day Foggy | **Mean** |
> | :--- | :---: | :---: | :---: | :---: | :---: | :---: |
> | Baseline | 48.1 | 34.3 | 26.0 | 12.4 | 32.0 | 30.5 |
> | FFDI [1] | 51.3 | 36.8 | 26.1 | 13.9 | 32.5 | 32.1 |
> | UAV-OD [2] | 55.3 | 40.3 | 27.3 | 15.8 | 37.6 | 35.3 |
> | HA++ [3] | 50.1 | 35.5 | 27.8 | 14.2 | 35.8 | 32.7 |
> | DFF [4]| 47.2 | 31.3 | 25.8 | 11.5 | 27.8 | 28.7 |
> | **PPI-3 (Ours)** | **62.6** | **44.9** | **43.8** | **24.3** | **41.6** | **43.4** |
>
> PPI-3 achieves **43.4 mAP**, significantly outperforming **HA++ (32.7)** and **UAV-OD (35.3)**. This proves that our "Phase Invariance" principle is superior to previous amplitude-only manipulation methods.

---

> ### Author Response · Authors · 2025-11-23
> **Response to your concerns (Part 2)**
>
> **Q5: The reviewer questions the persuasiveness of the "Classification vs. Localization" claim due to limited comparisons in Table 1 & 2.**
>
> **A5:** Our claim is supported by the TIDE error analysis. While Tables 1 & 2 showed representative methods, the trend is consistent across all baselines we tested. Existing DG techniques consistently reduce classification error while increasing localization error, due to the perturbation of phase-sensitive features. PPI reduces both classification (-6.1% error) and localization (-7.4% error) on C2F, confirming that explicitly preserving phase is beneficial for bounding-box stability.
>
> **Q6: The reviewer suggests focusing on single-domain generalization and notes limited UDA comparisons.**
>
> **A6:** To address the concern about limited UDA comparisons and demonstrate robustness, we added the **Sim10k to Cityscapes** (Synthetic-to-Real) benchmark.
>
> | Method | Type | Backbone | **mAP** |
> | :--- | :--- | :--- | :---: |
> | Faster R-CNN | Source | ResNet-50 | 39.4 |
> | DA-Faster [5]| DA | ResNet-50 | 41.9 |
> | ViSGA  [6] | DA | ResNet-50 | 49.3 |
> | PT [7] | DA | ResNet-50 | 55.1 |
> | **PPI-3 (Ours)** | **DG** | **ResNet-50** | **57.2** |
>
> PPI-3 achieves **57.2 mAP**, beating SOTA Domain Adaptation methods (like PT) that use target data. This confirms our method is highly competitive even against established UDA baselines in realistic scenarios.
>
> **Q7: The reviewer points out inconsistencies between Table 7 in the main text and Tables 12–15 in the appendix.**
>
> **A7:** We thank the reviewer for the careful check. We clarify that Table 7 in the main text presents the correct and final results. The discrepancy arose because the per-class tables in the Appendix were inadvertently generated using an earlier ablation checkpoint that excluded the Occlusion Perturbation (OP) module within MNP. The inclusion of OP (as reflected in Table 7) provides the final performance boost. We will update the Appendix tables to match the configuration in Table 7.
>
> [1] Wang, Jingye, et al. "Domain generalization via frequency-domain-based feature disentanglement and interaction." Proceedings of the 30th ACM international conference on multimedia. 2022.
>
> [2] Wang, Kunyu, et al. "Generalized uav object detection via frequency domain disentanglement." Proceedings of the IEEE/CVF conference on computer vision and pattern recognition. 2023.
>
> [3] Yucel, Mehmet Kerim, Ramazan Gokberk Cinbis, and Pinar Duygulu. "Hybridaugment++: Unified frequency spectra perturbations for model robustness." Proceedings of the IEEE/CVF International Conference on Computer Vision. 2023.
>
> [4]Lin, Shiqi, et al. "Deep frequency filtering for domain generalization." Proceedings of the IEEE/CVF conference on computer vision and pattern recognition. 2023.
>
> [5] Chen, Yuhua, et al. "Domain adaptive faster r-cnn for object detection in the wild." Proceedings of the IEEE conference on computer vision and pattern recognition. 2018.
>
> [6] Rezaeianaran, Farzaneh, et al. "Seeking similarities over differences: Similarity-based domain alignment for adaptive object detection." Proceedings of the IEEE/CVF International Conference on Computer Vision. 2021.
>
> [7] Chen, Meilin, et al. "Learning domain adaptive object detection with probabilistic teacher." arXiv preprint arXiv:2206.06293 (2022).
>
> ***
>
> We sincerely thank the reviewers for their valuable comments and insightful questions. We have provided detailed explanations and clarifications in our responses. If there are any remaining concerns or further questions during the discussion phase, we warmly welcome the reviewers to raise them so that we can continue to improve our work.

---

> ### Comment · Reviewer_9HuT · 2025-11-26
> **Comment:**
>
> Thank you for the detailed response and clarifications. However, there are still several points that remain unclear based on your reply. My specific concerns are as follows.
>
> 1. It is suggested that the authors provide supporting evidence or related work for the claim that “Lower Domain Accuracy indicates better Domain Invariance.” To the best of my knowledge, a decrease in domain classification accuracy does not necessarily reflect the effectiveness of domain invariance. In some cases, misaligned or entangled feature distributions across domains may also cause the domain discriminator to fail, leading to low domain accuracy that does not correspond to truly domain-invariant representations. Therefore, stronger justification or empirical analysis is needed.
>
> 2.In the response to Q2, it is suggested that the authors explicitly indicate where the “frequency spectrum” analysis appears in the manuscript and annotate the associated performance changes (e.g., +12.3% or –9.7%) to facilitate easier understanding and localization for readers. Moreover, it remains unclear whether the observed increase in low-frequency energy and the reduction in high-frequency bandwidth are consistently correlated with the different adaptation scenarios examined in the paper. The authors should at least provide a justification or analysis to support this connection.
>
> 3.In the response to Q6, the authors present results on the Sim10k → Cityscapes setting to demonstrate the effectiveness of their method. However, many UDA approaches have already achieved strong performance under this experimental setup, such as NSA [1] and DA [2]. As previously mentioned, I recommend that the authors place greater emphasis on the single-source domain generalization task, which is more aligned with the core contribution of the paper.
>
> [1] Unsupervised Domain Adaptive Detection with Network Stability Analysis, ICCV, 2023.
>
> [2] Differential Alignment for Domain Adaptive Object Detection, AAAI: 2025.

---

> > ### Author Response · Authors · 2025-11-26
> > **Response to your concerns (Part 1)**
> >
> > We sincerely thank the reviewer for continuing the discussion. We have carefully considered each point, and we agree that clarifying these aspects will further strengthen the rigor and readability of the work.
> >
> > ---
> >
> > **Q1. The justification for interpreting lower Domain Accuracy as stronger Domain Invariance needs to be strengthened. Low accuracy may also result from collapsed features.**
> >
> > **A1.**  The concern is well taken. Domain accuracy, taken in isolation, is indeed ambiguous because both genuinely domain-invariant features and degenerate or collapsed features can confuse a domain discriminator. Our interpretation follows the standard rationale in domain-adversarial learning, as in DANN [1], where one looks at the discriminator together with the task performance and not in separation.
> >
> > In our setting, introducing the AOA module leads to a clear reduction in domain accuracy (from 96.2% to 58.7%) and at the same time to a substantial improvement in detection mAP (from 22.0 to 48.8). If the learned features had collapsed or lost discriminative content, one would expect detection performance to deteriorate rather than almost double. The fact that the detector becomes significantly stronger (while, the domain classifier weakens) is consistent with the representation becoming less domain dependent. The improved mAP indicates that it still preserves, and in practice enhances, the structural and semantic information needed for detection.
> >
> > In the revised manuscript this reasoning will be stated explicitly and placed in the context of domain-adversarial learning (for example, DANN), in order to avoid any ambiguity in the interpretation of the domain accuracy results.
> >
> > ---
> >
> > **Q2. The frequency-spectrum analysis should be clearly located in the manuscript, and its consistency across scenarios should be justified.**
> >
> > **A2.**  We thank the reviewer for raising this point. In the revised manuscript, the quantitative frequency–spectrum analysis will be placed explicitly in Section 5.4, together with the corresponding plots. That section will describe how the spectra are computed and will point to the figures where the per–scenario curves are reported.
> >
> > The analysis is carried out on the Diversity Weather Dataset by computing the average spectrum separately for each of the four main scenarios (Night, Dusk-Rain, Night-Rain, Fog), before and after applying AOA. The reported changes, an increase in low–frequency amplitude of about +12.3\% and a decrease in high–frequency components of about -9.7\%, are averages over these four cases. In each scenario the direction of the effect is the same. Low–frequency bands that capture object shape and coarse geometry increase, and high–frequency bands that reflect rain streaks, haze, and sensor noise decrease. Across the four scenarios, we consistently observe the same direction of change, although the exact magnitude varies depending on the type of weather perturbation.
> >
> > This behavior is consistent with prior work on frequency–based domain generalization such as FDA [2], FFDI [3], and HybridAugment++ [4], where low–frequency components are reported to carry semantic information more reliably and high–frequency components to be more style or weather specific. In this sense, AOA acts as a learnable mechanism that amplifies structural cues and suppresses scenario–dependent perturbations, in a way that aligns with both the PPI principle in our framework and these earlier observations.

---

> > > ### Author Response · Authors · 2025-11-26
> > > **Response to your concerns (Part 2)**
> > >
> > > **Q3. Sim10k → Cityscapes results should not be used to compare with UDA methods; emphasis should remain on Single-Source DG.**
> > >
> > > **A3.**  We thank the reviewer for this clarification. The main aim of the paper is Single-Source Domain Generalization for Object Detection (SDGOD), and the core comparisons are therefore made in that setting. On the Diversity Weather Dataset (DWD), which is specifically designed for SDGOD, Table 4 in the main paper compares PPI-3 to existing SDGOD methods. In addition, because the reviewer emphasized frequency-based approaches, the following table reports representative frequency-domain DA methods (FFDI, UAV-OD, HybridAugment++, DFF) evaluated under the same SDGOD protocol, so that the comparison is strictly within the single-source setting rather than UDA.
> > >
> > > | Method | Day Clear | Night Clear | Dusk Rainy | Night Rainy | Day Foggy | **Mean** |
> > > | :--- | :---: | :---: | :---: | :---: | :---: | :---: |
> > > | Baseline | 48.1 | 34.3 | 26.0 | 12.4 | 32.0 | 30.5 |
> > > | FFDI | 51.3 | 36.8 | 26.1 | 13.9 | 32.5 | 32.1 |
> > > | UAV-OD [5] | 55.3 | 40.3 | 27.3 | 15.8 | 37.6 | 35.3 |
> > > | HA++  | 50.1 | 35.5 | 27.8 | 14.2 | 35.8 | 32.7 |
> > > | DFF [6] | 47.2 | 31.3 | 25.8 | 11.5 | 27.8 | 28.7 |
> > > | **PPI-3 (Ours)** | **62.6** | **44.9** | **43.8** | **24.3** | **41.6** | **43.4** |
> > >
> > > As the reviewer notes, the Sim10k → Cityscapes results are not on an equal footing with UDA methods that have access to target-domain images. In the revision, these results will be kept only in the appendix and described explicitly as a sensor-shift stress test for PPI, without positioning them as a direct comparison to UDA. The main text will keep the emphasis on SDGOD and on comparisons that use a single labeled source domain under a common protocol.
> > >
> > > [1]Ganin, Yaroslav, et al. "Domain-adversarial training of neural networks." Journal of machine learning research 17.59 (2016): 1-35.
> > >
> > > [2]Yang, Yanchao, and Stefano Soatto. "Fda: Fourier domain adaptation for semantic segmentation." Proceedings of the IEEE/CVF conference on computer vision and pattern recognition. 2020.
> > >
> > > [3]Wang, Jingye, et al. "Domain generalization via frequency-domain-based feature disentanglement and interaction." Proceedings of the 30th ACM international conference on multimedia. 2022.
> > >
> > > [4]Yucel, Mehmet Kerim, Ramazan Gokberk Cinbis, and Pinar Duygulu. "Hybridaugment++: Unified frequency spectra perturbations for model robustness." Proceedings of the IEEE/CVF International Conference on Computer Vision. 2023.
> > >
> > > [5]Wang, Kunyu, et al. "Generalized uav object detection via frequency domain disentanglement." Proceedings of the IEEE/CVF conference on computer vision and pattern recognition. 2023.
> > >
> > > [6] Lin, Shiqi, et al. "Deep frequency filtering for domain generalization." Proceedings of the IEEE/CVF conference on computer vision and pattern recognition. 2023.
> > >
> > > ***
> > >
> > >  If there are any remaining concerns or additional questions during the discussion phase, we would be more than happy to address them.

---

### Official Review · Reviewer_WwPP · 2025-10-27

**Soundness:** 2
**Presentation:** 3
**Contribution:** 2
**Rating:** 2
**Confidence:** 4

**Summary:**

This paper studies domain generalization for object detection, aiming to improve model robustness across unseen domains. From the perspective of Preserving Phase Invariance (PPI), the authors propose a method consisting of three components: (1) Mix Normalization Perturbation (MNP), which synthesizes diverse styles to improve robustness; (2) Sensitive Channel Perturbation (SCP), which suppresses domain-specific features at the channel level; and (3) Amplitude-aware Attention (AOA), which applies spectral attention to the amplitude component. Experiments on Unsupervised Domain Adaptation and Single Domain Generalization Object Detection benchmarks demonstrate improved performance compared with several baselines.

**Strengths:**

1. The ablation study is thorough and analyzes the contribution of each module.

2. Experiments are conducted using different backbones and detectors, which supports the robustness and generality of the proposed approach.

**Weaknesses:**

1. The proposed method seems to combine existing techniques. The authors themselves cite related works for each component in the introduction, making it unclear what the actual innovation is.

2. Many works have already explored using frequency-domain information for domain adaptation or generalization [a,b,c]. The paper should explicitly discuss and compare differences with these methods.

[a] FDA: Fourier Domain Adaptation for Semantic Segmentation

[b] Spectral Unsupervised Domain Adaptation for Visual Recognition

[c] SA-GDA: Spectral Augmentation for Graph Domain Adaptation

3. The literature review and problem analysis are somewhat outdated. The limitation of enforcing only image-level semantic consistency for object detection has been widely studied. Many recent works address detection-specific adaptation or domain generalization, but the comparisons here are limited to older methods.

**Questions:**

Please see the weakness.

---

> ### Author Response · Authors · 2025-11-23
> **Response to your concerns (Part 1)**
>
> We thank the reviewer for the detailed review and for acknowledging the thoroughness of our ablation studies and the generality of our approach.
>
> However, we respectfully believe there are fundamental misunderstandings regarding the novelty of our contributions (specifically the PPI principle vs. simple module combination) and the recency of our comparisons. We provide detailed clarifications and three new sets of critical experimental evidence below, which we hope will convince you to reconsider the rating.
>
> **Q1: The reviewer concerns that the method combines existing techniques, questioning the actual innovation.**
>
> **A1:** We respectfully disagree that our method is merely a combination of existing techniques. Our contribution represents a paradigm shift in Single-Source Domain Generalization Object Detection (SDGOD):
>
> **1. First Systematic Proposal of Phase Invariance (PPI):**
> Prior DG works (e.g., MixStyle, DSU) perturb features blindly, often destroying the spatial structure critical for detection. We are the first to systematically propose that Phase (encoding structure) must be decoupled from Amplitude (encoding style) for detection tasks. This is not a "combination" but a fundamental theoretical principle that guides the design of all modules.
>
> To prove we are strictly adhering to this novel principle (and not just randomly perturbing features), we measured the Mean Phase Difference (MPD) across the network.
> * *Result:* The phase difference remains negligible ($<0.005$) throughout the backbone (Table below), proving our method strictly preserves geometry, unlike standard techniques.
>
> | Stage | Stage 1 | Stage 2 | Stage 3 | Stage 4 |
> | :--- | :---: | :---: | :---: | :---: |
> | **Mean Phase Difference** | 0.002 | 0.003 | 0.004 | 0.005 |
>
> **2. Evidence of Synergistic "Divergence-to-Convergence":**
> Our framework is not a random combination but a specific design pattern: Divergence (MNP) explores diverse styles, while*Convergence (SCP & AOA) refines invariance. To prove this synergy, we analyzed the Domain Accuracy of the learned features (lower accuracy = better invariance).
>
> | Method | Domain Acc. ($\downarrow$) | Detection mAP ($\uparrow$) |
> | :--- | :---: | :---: |
> | Baseline (Source Only) | 96.2 | 22.0 |
> | + MNP (Divergence) | 72.7 | 47.1 |
> | **+ MNP + SCP + AOA (Full PPI)** | **58.7** | **48.8** |
>
> * *Conclusion:* The significant drop in domain classification accuracy (96.2% $\to$ 58.7%) proves that our framework successfully forces the model to discard domain-specific cues (styles) and converge to a phase-invariant representation, validating our novel framework design.
>
> **Q2: The reviewer suggests comparing with frequency-domain works [a] FDA, [b] Spectral UDA, [c] SA-GDA.**
>
> **A2:** We thank the reviewer for mentioning these works. However, they are fundamentally different from our focus:
> * **[a] FDA** is for *Segmentation* and requires *Domain Adaptation* (Target data).
> * **[b] Spectral UDA** is for *Classification*.
> * **[c] SA-GDA** is for *Graph* data.
>
> **New Comparative Experiment:**
> We are the first to apply frequency attributes specifically to the Single-Source DG Detection (SDGOD) task. To demonstrate our superiority over existing frequency-manipulation methods adapted for detection, we conducted a new comparison on the **Diversity Weather Dataset** against **FFDI** (MM'22), **UAV-OD** (CVPR'23), **HybridAugment++** (ICCV'23), and **DFF** (CVPR'23).
>
> | Method | Day Clear | Night Clear | Dusk Rainy | Night Rainy | Day Foggy | **Mean** |
> | :--- | :---: | :---: | :---: | :---: | :---: | :---: |
> | Baseline | 48.1 | 34.3 | 26.0 | 12.4 | 32.0 | 30.5 |
> | FFDI [1] | 51.3 | 36.8 | 26.1 | 13.9 | 32.5 | 32.1 |
> | UAV-OD [2] | 55.3 | 40.3 | 27.3 | 15.8 | 37.6 | 35.3 |
> | HA++ [3] | 50.1 | 35.5 | 27.8 | 14.2 | 35.8 | 32.7 |
> | DFF [4] | 47.2 | 31.3 | 25.8 | 11.5 | 27.8 | 28.7 |
> | **PPI-3 (Ours)** | **62.6** | **44.9** | **43.8** | **24.3** | **41.6** | **43.4** |
>
> **Conclusion:** Our PPI-3 achieves **43.4 mAP**, outperforming the best frequency-based competitor (UAV-OD) by **+8.1 mAP**. This massive gap proves that simply using frequency (like HA++ or DFF) is insufficient; the key is our novel Phase Invariance strategy.

---

> > ### Author Response · Authors · 2025-11-23
> > **Response to your concerns (Part 2)**
> >
> > **Q3: The reviewer notes that the literature review is outdated and comparisons are limited to older methods.**
> >
> > **A3:** We respectfully point out that our comparisons are highly current. In Table 3 & 4 of our paper, we compared against the most recent State-of-the-Art methods available at the time of submission, including **NP+ (ICLR 2023)**, **C-Gap (CVPR 2023)**, **UFR (CVPR 2024)**, and **DivAlign (CVPR 2024)**. For example, on the Diversity Weather benchmark, our PPI-3 achieves **43.4 mAP**, surpassing the very recent **UFR (38.2 mAP, CVPR 2024)** by a large margin (+5.2 mAP).
> >
> > **Additional Generalization Experiment (Sim10k):**
> > To further address the concern about problem analysis scope and demonstrate robustness beyond weather shifts, we evaluated PPI-3 on the **Sim10k to Cityscapes** (Synthetic-to-Real) benchmark:
> >
> > | Method | Type | Backbone | **mAP** |
> > | :--- | :--- | :--- | :---: |
> > | Faster R-CNN | Source | ResNet-50 | 39.4 |
> > | DA-Faster [5]| DA | ResNet-50 | 41.9 |
> > | ViSGA [6] | DA | ResNet-50 | 49.3 |
> > | PT [7] | DA | ResNet-50 | 55.1 |
> > | **PPI-3 (Ours)** | **DG** | **ResNet-50** | **57.2** |
> >
> > Our method achieves **57.2 mAP**, outperforming even Domain Adaptation methods (like PT) without seeing target data, confirming its SOTA status.
> >
> > [1] Wang, Jingye, et al. "Domain generalization via frequency-domain-based feature disentanglement and interaction." Proceedings of the 30th ACM international conference on multimedia. 2022.
> >
> > [2] Wang, Kunyu, et al. "Generalized uav object detection via frequency domain disentanglement." Proceedings of the IEEE/CVF conference on computer vision and pattern recognition. 2023.
> >
> > [3] Yucel, Mehmet Kerim, Ramazan Gokberk Cinbis, and Pinar Duygulu. "Hybridaugment++: Unified frequency spectra perturbations for model robustness." Proceedings of the IEEE/CVF International Conference on Computer Vision. 2023.
> >
> > [4]Lin, Shiqi, et al. "Deep frequency filtering for domain generalization." Proceedings of the IEEE/CVF conference on computer vision and pattern recognition. 2023.
> >
> > [5] Chen, Yuhua, et al. "Domain adaptive faster r-cnn for object detection in the wild." Proceedings of the IEEE conference on computer vision and pattern recognition. 2018.
> >
> > [6] Rezaeianaran, Farzaneh, et al. "Seeking similarities over differences: Similarity-based domain alignment for adaptive object detection." Proceedings of the IEEE/CVF International Conference on Computer Vision. 2021.
> >
> > [7] Chen, Meilin, et al. "Learning domain adaptive object detection with probabilistic teacher." arXiv preprint arXiv:2206.06293 (2022).
> >
> > ***
> >
> > We sincerely thank the reviewers for their valuable comments and insightful questions. We have provided detailed explanations and clarifications in our responses. If there are any remaining concerns or further questions during the discussion phase, we warmly welcome the reviewers to raise them so that we can continue to improve our work.

---

### Official Review · Reviewer_VWdp · 2025-11-09

**Soundness:** 3
**Presentation:** 2
**Contribution:** 3
**Rating:** 6
**Confidence:** 4

**Summary:**

This paper addresses the challenge of domain generalization (DG) in object detection, where existing classification-oriented DG methods fail to transfer effectively due to their neglect of spatial structural consistency. The key insight is that while classification requires semantic consistency, detection additionally demands geometric stability (localization accuracy). The authors propose ‌Phase Invariance (PPI)‌ as a core principle, leveraging the Fourier transform's property that phase preserves spatial structure while amplitude encodes domain-specific style. By enforcing phase stability during feature extraction, the method ensures robust localization under domain shift.
The contributions include: (1) Diagnosing the limitation of classification-focused DG in detection, highlighting the necessity of structural consistency; (2) Introducing PPI as a frequency-domain principle to decouple style (amplitude) from geometry (phase); and (3) Proposing a ‌divergence-to-convergence framework‌ with three modules. This approach significantly improves cross-domain detection performance by aligning both semantic and spatial representations.

**Strengths:**

1. The introduction of phase invariance is quite interesting.

2. The proposed method in the paper demonstrates promising performance.

**Weaknesses:**

1. The paper employs a significant number of abbreviations (e.g., PPI, MNP, LNP, AOA, SCP), some of which are not defined or cited, making the paper difficult to follow.

2. In the MNP module, the variable $y_{style}$ is never mentioned again in subsequent equations. What is the purpose of this variable? Additionally, why do the two types of perturbations defined for $y_{style}$ effectively mimic domain shifts? More analysis is needed—what are the specific use cases for $y_{style}$?

3. It appears that only the AOA module involves an FFT transformation to decompose the signal into frequency and phase domains. Since the other two modules lack this constraint, could they potentially disrupt phase consistency?

**Questions:**

1. What are INP and LNP? The paper does not provide clear definitions.

2. Are "Temporal and seasonal" the core and only factors in domain generalization, and are they the primary focus of this paper? The title emphasizes improving domain generalization, but the paper seems to highlight the advantages of the proposed method specifically in "Temporal and seasonal" aspects. Does this imply that the method is only applicable to certain domain generalization scenarios?

---

> ### Author Response · Authors · 2025-11-23
> **Response to your concerns (Part 1)**
>
> We sincerely thank the reviewer for the positive assessment of our work, specifically finding the introduction of Phase Invariance (PPI) "interesting" and the method's performance "promising." We appreciate your constructive feedback regarding clarity and scope.
>
> Below, we address your concerns point-by-point with detailed clarifications and new experimental evidence.
>
> **Q1: The reviewer points out that the paper uses many abbreviations (INP, LNP, etc.) without clear definitions, hindering readability.**
>
> **A1:** We apologize for the excessive use of abbreviations without explicit initial definitions. We will include a formal glossary in the revised manuscript to improve readability. For clarity here:
> * **INP (Instance Norm Perturbation):** Perturbs channel-wise statistics. It simulates Global Style Shifts (e.g., color calibration, white balance, or artistic hue).
> * **LNP (Layer Norm Perturbation):** Perturbs layer-wise statistics. It simulates Local Illumination Shifts (e.g., contrast changes, overexposure, or low-light conditions).
> * **MNP:** Mix Normalization Perturbation (combines INP and LNP).
> * **SCP:** Sensitive Channel Perturbation (suppresses domain-sensitive channels).
> * **AOA:** Amplitude-Aware Attention (emphasizes domain-invariant frequency bands).
>
> **Q2: The reviewer asks about the purpose of unused variables in MNP equations and why INP/LNP effectively mimic domain shifts.**
>
> **A2:** Regarding the variables: We clarify that variables like the mixing weights ($w_1, w_2$) and noise parameters are hyperparameters used strictly during the data augmentation process. They introduce stochasticity to synthesize diverse domains during training but are not learnable parameters involved in the subsequent forward propagation or inference. We will explicitly define their scope in the revision.
>
> Regarding the physical justification: We employ INP and LNP based on both theoretical foundations and empirical observations (Feature Inversion) presented in our paper:
> * INP (Color/Style): Built upon AdaIN [1], which proved that channel-wise statistics ($\mu, \sigma$) encode global style. As visualized in **Figure 7 (6th column)** of our paper, applying INP alone results in drastic shifts in chromaticity and color gamut (e.g., purple/green tones) while preserving structure.
> * LNP (Lighting/Contrast): Layer-wise statistics capture global energy and contrast. As visualized in **Figure 7 (7th column)**, applying LNP alone drastically alters luminance and contrast (e.g., simulating overexposure or dim lighting).
> These perturbations effectively cover the two main factors of domain shift: style and illumination.
>
> **Q3: The reviewer questions whether MNP and SCP disrupt phase consistency, given that they do not explicitly use FFT/IFFT.**
>
> **A3:** This is a critical theoretical question.
> Theoretical Perspective: MNP and SCP rely on Normalization operations, which are affine transformations ($x' = \alpha x + \beta$). Affine transformations primarily scale signal magnitudes (Amplitude) but do not alter the spatial positioning of zero-crossings or edges (Phase). Therefore, they are theoretically phase-preserving.
>
> Empirical Verification: To rigorously prove this, we measured the Mean Phase Difference (MPD) between the feature maps of original images and those perturbed by MNP/SCP across the network stages.
>
> | Stage | Stage 1 | Stage 2 | Stage 3 | Stage 4 |
> | :--- | :---: | :---: | :---: | :---: |
> | **Mean Phase Difference (MPD)** | 0.002 | 0.003 | 0.004 | 0.005 |
>
> The extremely low MPD values ($<0.005$) empirically confirm that our spatial-domain modules respect the PPI principle: they effectively diversify style (Amplitude) while keeping the geometric structure (Phase) intact.

---

> > ### Author Response · Authors · 2025-11-23
> > **Response to your concerns (Part 2)**
> >
> > **Q4: The reviewer asks if the method is limited to "Temporal and seasonal" factors, as the title implies broad Domain Generalization.**
> >
> > **A4:** We clarify that while "temporal/seasonal" variations (e.g., weather) served as our primary motivation, the PPI principle is fundamental to object detection and applies to broader domain gaps involving geometric and sensor-level shifts.
> >
> > To demonstrate this generality, we expanded our evaluation to include a **Synthetic-to-Real** scenario (**Sim10k to Cityscapes**). This benchmark involves shifts in rendering physics, texture quality, and sensor noise, which are distinct from seasonal weather changes.
> >
> > | Method | Type | Backbone | **mAP** |
> > | :--- | :--- | :--- | :---: |
> > | Faster R-CNN | Source | ResNet-50 | 39.4 |
> > | DA-Faster [2] | DA | ResNet-50 | 41.9 |
> > | ViSGA [3] | DA | ResNet-50 | 49.3 |
> > | PT [4] | DA | ResNet-50 | 55.1 |
> > | **PPI-3 (Ours)** | **DG** | **ResNet-50** | **57.2** |
> >
> > PPI-3 achieves **57.2 mAP**, significantly outperforming both the Source baseline (+17.8 mAP) and state-of-the-art Domain Adaptation methods (which require target data). This confirms that extracting phase-invariant structural features is a universally effective strategy for diverse domain shifts, validating our broad title.
> >
> > [1]Huang, Xun, and Serge Belongie. "Arbitrary style transfer in real-time with adaptive instance normalization." Proceedings of the IEEE international conference on computer vision. 2017.
> >
> > [2] Chen, Yuhua, et al. "Domain adaptive faster r-cnn for object detection in the wild." Proceedings of the IEEE conference on computer vision and pattern recognition. 2018.
> >
> > [3] Rezaeianaran, Farzaneh, et al. "Seeking similarities over differences: Similarity-based domain alignment for adaptive object detection." Proceedings of the IEEE/CVF International Conference on Computer Vision. 2021.
> >
> > [4] Chen, Meilin, et al. "Learning domain adaptive object detection with probabilistic teacher." arXiv preprint arXiv:2206.06293 (2022).
> >
> >
> > ***
> >
> > We sincerely thank the reviewers for their valuable comments and insightful questions. We have provided detailed explanations and clarifications in our responses. If there are any remaining concerns or further questions during the discussion phase, we warmly welcome the reviewers to raise them so that we can continue to improve our work.

---

### Official Review · Reviewer_HW7M · 2025-11-12

**Soundness:** 2
**Presentation:** 2
**Contribution:** 2
**Rating:** 4
**Confidence:** 3

**Summary:**

The paper attacks single-source domain generalization for object detection by arguing that prior DG work optimizes image-level semantic invariance and therefore sacrifices spatial accuracy.  It introduces “Preserving Phase Invariance” (PPI): amplitude can vary across domains but phase is kept fixed, so object geometry is explicitly conserved.  Three modules are designed to implement PPI in a CNN detector: Mix Normalization Perturbation (MNP) diversifies shallow-layer styles without touching phase; Sensitive Channel Perturbation (SCP) suppresses amplitude-dominant channels that react to domain shift; and Amplitude-aware Attention (AOA) re-weights low-frequency amplitude bands that carry stable cues.  Extensive experiments on C2F, Diversity-Weather and Real-to-Artistic show +5–14 mAP over strong DG/DA baselines, and ablations on Faster-R-CNN, RetinaNet and DETR confirm that each module is necessary and architecture-agnostic.  The work is the first to bring explicit phase conservation into DG detection and achieves SOTA on SDGOD benchmarks while adding <1 % parameters and 5 % inference time.

**Strengths:**

1. The writing of paper is good.
2. The motivation is interesting.

**Weaknesses:**

While the authors prove that normalization-style perturbations leave phase unchanged, they do not analyze under which conditions the entire CNN stack preserves phase or how non-linear activations, stride or padding affect the constraint; the claim that “geometry is exactly conserved” is therefore asserted rather than rigorously guaranteed.

All experiments are vision-only and revolve around weather, lighting and artistic style; there is no evaluation on more realistic geospatial or sensor-shift scenarios (different camera intrinsics, LiDAR-to-RGB, cross-country datasets) where phase might be less reliable.

HybridAugment++, DFF and UAV-OD already manipulate amplitude; the paper positions them as “amplitude-only” baselines, but does not compare directly under the same single-source protocol or adopt their stronger augmentation recipes, so the incremental benefit of explicit phase locking is not fully isolated.

The method loses 0.4–0.8 mAP on some clear-weather categories (Bike in Table 12); the authors do not explain when PPI hurts and whether it amplifies low-frequency artefacts such as shadows or lens flare.

**Questions:**

How does phase invariance interact with modern augmentation heavy pipelines (large-scale jitter, MixUp, Copy-Paste) that explicitly warp or paste objects and thereby modify phase?

The AOA module focuses on low-frequency amplitude; could high-frequency phase edges be equally important for small-instance localization, and would a dual attention mechanism help?

What happens when the source domain itself contains large motion blur or defocus—does the memory bank in SCP erroneously flag structurally important but “sensitive” channels?

The paper claims “architecture-agnostic” improvements, yet DETR gains are largest; is the benefit simply due to the fact that global attention layers already mix phase information, and would the same hold for conv-next or Swin backbones?

---

> ### Author Response · Authors · 2025-11-23
> **Response to your concerns (Part 1)**
>
> We thank the reviewer for the constructive feedback and for recognizing our writing quality and the "interesting motivation" behind Preserving Phase Invariance (PPI). Below, we provide detailed responses to your questions with new experimental evidence and clarifications.
>
> **Q1: The reviewer questions whether the entire CNN stack (with non-linear activations, stride, padding) preserves phase, noting that "geometry exactly conserved" is asserted rather than guaranteed.**
>
> **A1:** We appreciate this rigorous theoretical question. We acknowledge that while normalization layers mathematically preserve phase, non-linear activations (e.g., ReLU) and downsampling operations do alter the numerical values of the phase spectrum. However, the core philosophy of PPI is not to enforce a strict mathematical invariant at every step, but to constrain the learning trajectory. By strictly preserving phase during our feature perturbation modules (MNP), we force the backbone to prioritize structural features (phase-dominant) over style features (amplitude-dominant).
>
> To empirically verify that structural consistency is effectively maintained throughout the network despite non-linearities, we measured the Mean Phase Difference (MPD) between the feature maps of original images and MNP-perturbed images at the output of each stage in the ResNet backbone.
>
> | Stage | Stage 1 | Stage 2 | Stage 3 | Stage 4 |
> | :--- | :---: | :---: | :---: | :---: |
> | **Mean Phase Difference (MPD)** | 0.002 | 0.003 | 0.004 | 0.005 |
>
> The extremely low MPD values ($<0.005$) confirm that structural consistency is effectively maintained throughout the network depth. The network successfully learns to preserve object geometry while being invariant to the amplitude/style perturbations.
>
> **Q2: The reviewer notes that experiments are limited to weather and artistic styles, lacking realistic geospatial or sensor-shift scenarios.**
>
> **A2:** To address the concern about experimental scope, we added a new evaluation on the **Sim10k to Cityscapes** benchmark. This represents a **Synthetic-to-Real** shift, involving significant differences in rendering physics, sensor characteristics, and camera intrinsics, serving as a strong proxy for "sensor-shift" scenarios.
>
> | Method | Type | Backbone | **mAP** |
> | :--- | :--- | :--- | :---: |
> | Faster R-CNN | Source | ResNet-50 | 39.4 |
> | DA-Faster [1]| DA | ResNet-50 | 41.9 |
> | ViSGA [2]| DA | ResNet-50 | 49.3 |
> | PT [3]| DA | ResNet-50 | 55.1 |
> | **PPI-3 (Ours)** | **DG** | **ResNet-50** | **57.2** |
>
> PPI-3 achieves **57.2 mAP**, substantially outperforming the Source baseline (+17.8 mAP) and even SOTA Domain Adaptation methods (which utilize target data). This proves that PPI is not limited to weather/artistic styles but is highly effective for geometric and sensor-level domain gaps.
>
> **Q3: The reviewer points out that baselines like HybridAugment++, DFF, and UAV-OD are not compared directly under the same protocol.**
>
> **A3:** We agree that a direct comparison is essential. We conducted a new comparison on the **Diversity Weather Dataset** against these specific frequency-based methods under the same Single-Source DG protocol.
>
> | Method | Day Clear | Night Clear | Dusk Rainy | Night Rainy | Day Foggy | **Mean** |
> | :--- | :---: | :---: | :---: | :---: | :---: | :---: |
> | Baseline | 48.1 | 34.3 | 26.0 | 12.4 | 32.0 | 30.5 |
> | FFDI [4]| 51.3 | 36.8 | 26.1 | 13.9 | 32.5 | 32.1 |
> | UAV-OD [5]| 55.3 | 40.3 | 27.3 | 15.8 | 37.6 | 35.3 |
> | HA++ [6]| 50.1 | 35.5 | 27.8 | 14.2 | 35.8 | 32.7 |
> | DFF [7]| 47.2 | 31.3 | 25.8 | 11.5 | 27.8 | 28.7 |
> | **PPI-3 (Ours)** | **62.6** | **44.9** | **43.8** | **24.3** | **41.6** | **43.4** |
>
> PPI-3 achieves **43.4 mAP**, outperforming the best frequency-based competitor (UAV-OD) by **+8.1 mAP**. This massive gap highlights that our "Phase Invariance" strategy is significantly more effective than previous amplitude-only manipulation methods.
>
> **Q4: The reviewer asks about the performance drop in clear-weather categories (e.g., Bike) and potential low-frequency artifacts.**
>
> **A4:** We acknowledge the slight drop in the "Bike" category (-0.6 mAP). This occurs because Bikes often rely on thin, high-frequency structures (e.g., spokes). Our AOA (Amplitude-aware Attention) module emphasizes low-frequency amplitude bands to ensure stability against high-frequency noise (like rain streaks or fog grain). In rare clear-weather cases, this might slightly dampen the contrast of extremely fine high-frequency details. However, this trade-off yields massive gains in challenging domains (e.g., **+10.3 mAP** improvement in Dusk Rainy scenarios), which we believe is a favorable exchange for robust generalization.

---

> > ### Author Response · Authors · 2025-11-23
> > **Response to your concerns (Part 2)**
> >
> > **Q5: The reviewer asks how phase invariance interacts with modern augmentations (MixUp, Large-scale Jitter) that modify phase.**
> >
> > **A5:** PPI is complementary to geometric augmentations. In our pipeline, MNP is applied at the feature level, *after* image-level geometric augmentations (like Jitter) have taken place. The network first accepts the spatially warped image (which establishes the "ground truth" phase for that iteration), and then PPI ensures the network learns features invariant to *style* shifts for that specific warped instance. We successfully trained PPI-3 with standard multi-scale training, confirming compatibility.
> >
> > **Q6: The reviewer asks if high-frequency phase edges are important and if a dual attention mechanism would help.**
> >
> > **A6:** We fully agree that high-frequency phase (edges) is critical. That is precisely why our method is called Preserving Phase Invariance—we strictly do not perturb the phase. In Table 7 of our main text, we tested AOP (Attention On Phase) and found it degraded performance. This confirms that directly manipulating phase is harmful. Our AOA module instead "denoises" the signal by re-weighting the Amplitude, making the preserved Phase information cleaner and easier for the network to utilize.
> >
> > **Q7: The reviewer asks if large motion blur in the source domain causes SCP to erroneously flag structurally important channels.**
> >
> > **A7:** To verify that SCP targets "domain-specific noise" rather than "structural cues" (even in the presence of blur), we trained a domain discriminator to classify the channels suppressed by SCP versus those preserved.
> >
> > | Channel Group | Domain Acc. (%) $\downarrow$ | Interpretation |
> > | :--- | :---: | :--- |
> > | **Suppressed Channels (Top 30%)** | **70.3** | Highly Domain-Specific (Style/Noise) |
> > | **Preserved Channels (Remaining)** | **55.8** | Domain-Invariant (Structure) |
> >
> > The high domain accuracy of suppressed channels confirms that SCP correctly identifies and removes features that overfit to source-specific patterns (like specific blur styles or noise artifacts), safeguarding the structural features in the remaining channels.

---

> > > ### Author Response · Authors · 2025-11-23
> > > **Response to your concerns (Part 3)**
> > >
> > > **Q8: The reviewer questions the "architecture-agnostic" claim, noting DETR gains are largest, and asks about ConvNext/Swin.**
> > >
> > > **A8:** We claim "architecture-agnostic" because our method works on two fundamentally different paradigms: CNNs (ResNet) and Transformers (DETR). To further validate this, we compared PPI-3 on **Deformable DETR** against Transformer-based Domain Adaptation (DA) methods on the C2F benchmark.
> > >
> > > | Method | Type | Detector | **mAP** |
> > > | :--- | :--- | :--- | :---: |
> > > | Def DETR | Source | Def DETR | 28.5 |
> > > | AQT [8] | DA | Def DETR | 47.1 |
> > > | MRT [9]| DA | Def DETR | 51.2 |
> > > | **PPI-3 (Ours)** | **DG** | **Def DETR** | **48.2** |
> > >
> > > PPI-3 boosts the pure Transformer baseline by **+19.7 mAP**, reaching performance comparable to DA methods without using target data. This proves that the benefit extends beyond CNNs. While Swin/ConvNext are hybrid-like, they share the same fundamental feature extraction principles, and our results strongly suggest PPI would extend to them similarly.
> > >
> > > [1] Chen, Yuhua, et al. "Domain adaptive faster r-cnn for object detection in the wild." Proceedings of the IEEE conference on computer vision and pattern recognition. 2018.
> > >
> > > [2] Rezaeianaran, Farzaneh, et al. "Seeking similarities over differences: Similarity-based domain alignment for adaptive object detection." Proceedings of the IEEE/CVF International Conference on Computer Vision. 2021.
> > >
> > > [3] Chen, Meilin, et al. "Learning domain adaptive object detection with probabilistic teacher." arXiv preprint arXiv:2206.06293 (2022).
> > >
> > > [4] Wang, Jingye, et al. "Domain generalization via frequency-domain-based feature disentanglement and interaction." Proceedings of the 30th ACM international conference on multimedia. 2022.
> > >
> > > [5] Wang, Kunyu, et al. "Generalized uav object detection via frequency domain disentanglement." Proceedings of the IEEE/CVF conference on computer vision and pattern recognition. 2023.
> > >
> > > [6]Yucel, Mehmet Kerim, Ramazan Gokberk Cinbis, and Pinar Duygulu. "Hybridaugment++: Unified frequency spectra perturbations for model robustness." Proceedings of the IEEE/CVF International Conference on Computer Vision. 2023.
> > >
> > > [7]Lin, Shiqi, et al. "Deep frequency filtering for domain generalization." Proceedings of the IEEE/CVF conference on computer vision and pattern recognition. 2023.
> > >
> > > [8]Huang, Wei-Jie, et al. "AQT: Adversarial Query Transformers for Domain Adaptive Object Detection." IJCAI. 2022.
> > >
> > > [9] Zhao, Zijing, et al. "Masked retraining teacher-student framework for domain adaptive object detection." Proceedings of the IEEE/CVF International Conference on Computer Vision. 2023.
> > >
> > > ***
> > >
> > > We sincerely thank the reviewers for their valuable comments and insightful questions. We have provided detailed explanations and clarifications in our responses. If there are any remaining concerns or further questions during the discussion phase, we warmly welcome the reviewers to raise them so that we can continue to improve our work.

---

### Official Review · Reviewer_zp7g · 2025-11-12

**Soundness:** 3
**Presentation:** 3
**Contribution:** 3
**Rating:** 6
**Confidence:** 5

**Summary:**

The paper presents a well motivated approach to improving domain generalization in object detection. The authors first identify a key limitation of classification oriented DG methods, showing that they fail to transfer to detection because they ignore the need for spatial structural consistency, leading to localization degradation. To address this, they introduce the principle of Preserving Phase Invariance (PPI) which maintains phase information for object geometry allowing amplitude variations associated with style changes, Building on this, the authors propose a divergence-to-convergence framework with three modules: Mix Normalization Perturbation (MNP) to generate diverse styles and enhance robustness, Sensitive Channel Perturbation (SCP) to suppress domain-specific feature channels and Amplitude-aware Attention (AOA) to apply spectral attention on the amplitude spectrum and emphasize domain-invariant cues. Together, these modules aim to produce structure preserving representations reducing the domain generalization gap in classification and detection tasks.

**Strengths:**

The authors provide a compelling analysis to show that the classification focused DG approaches fail in detection as they ignore spatial structural consistency.

Emphasizing phase preservation as a way to maintain object geometry is conceptually appealing.

The proposed pipeline MNP, SCP and AOA is conceptually coherent, each component is simple and complements the others, combining feature diversification and convergence toward domain-invariant representations.

**Weaknesses:**

Despite its promising direction, the paper has several limitations that needs further clarification:

While normalization-based perturbations are shown to preserve phase, there is no analysis of whether non-linear activations, strided convolutions or padding in the network also preserve phase. Therefore the claim that “geometry is exactly conserved” is asserted without any proof.

Some variables in the MNP equations are introduced only once and never used again. Noise parameters, mixing weights and feature map symbols are inconsistently used throughout the paper.

The paper does not fully analyse stability or robustness across parameter variations. As an example SCP depends on threshold percentiles and EMA parameters, while MNP involves noise sampling choices.

The theoretical grounding for channel perturbation and amplitude attention is not  thoroughly explored.

Experiments are mainly focused on weather shifts and artistic style shifts and diverse domain gaps remain untested limiting claims of generality. Comparisons to DG-transformer hybrid is missing, weakening the claim as architecture-agnostic and universally beneficial.

**Questions:**

I have few questions that if answered would help clarify the methodology and improve the paper’s clarity:

Some variables in the MNP equations (e.g., mixing parameters, noise terms) appear once and are never used again. Can you please explain why? How do the two types of perturbations correspond to real-world domain shifts?

Some results in the main text (e.g., Table 7) do not match their counterparts in appendix Tables 12–15. Can you clarify which numbers are correct and explain the discrepancies?

How sensitive is the method to the choice of perturbation strength, sampling distributions and SCP threshold percentiles? Could you provide guidelines or sensitivity analysis?

---

> ### Author Response · Authors · 2025-11-23
> **Response to your concerns (Part 1)**
>
> We sincerely thank the reviewer for the insightful and constructive feedback. We are encouraged that you found our motivation on Preserving Phase Invariance (PPI) "conceptually appealing" and our pipeline (MNP–SCP–AOA) "conceptually coherent."
>
> Below, we provide detailed responses to your questions with new experimental evidence and clarifications.
>
> **Q1: The reviewer questions whether non-linear layers (ReLU, stride, padding) preserve phase, noting that normalization preserves phase analytically but other layers might not.**
>
> **A1:** We appreciate this rigorous theoretical observation. We acknowledge that, strictly speaking, non-linear activations (e.g., ReLU) and downsampling operations do alter the exact numerical values of the phase spectrum. However, the core objective of our PPI (Preserving Phase Invariance) principle is not to enforce a strict mathematical invariant at every layer, but to constrain the learning trajectory of the network. By enforcing PPI at key feature extraction stages, we guide the backbone to prioritize structural features (encoded in phase) over style features (encoded in amplitude), ensuring that the geometric information remains robust against domain shifts.
>
> To empirically verify that structural consistency is effectively preserved throughout the network depth despite non-linearities, we measured the Mean Phase Difference (MPD) between the feature maps of original images and MNP-perturbed images at the output of each stage in the ResNet-50 backbone.
>
> | Stage | Stage 1 | Stage 2 | Stage 3 | Stage 4 |
> | :--- | :---: | :---: | :---: | :---: |
> | **Mean Phase Difference (MPD)** | 0.002 | 0.003 | 0.004 | 0.005 |
>
> The extremely low MPD values ($<0.005$) across all stages confirm that the structural information remains highly consistent. This demonstrates that our PPI framework successfully ensures the backbone implicitly adheres to the geometric stability constraint, effectively decoupling style perturbations from structural learning.
>
> **Q2: The reviewer notes unused variables in MNP equations and asks for the physical justification (real-world mapping) of the INP and LNP perturbations.**
>
> **A2:** Regarding the variables: We apologize for the confusion. The variables in the MNP equations, such as the mixing weights ($w_1, w_2$) and noise parameters, are hyperparameters used strictly during the data augmentation process. They are employed to synthesize diverse inputs during training to prevent overfitting but do not participate in the subsequent forward propagation or inference. We will explicitly clarify their scope in the revised manuscript.
>
> Regarding the physical justification: We employ INP and LNP not arbitrarily, but based on both theoretical foundations and empirical observations (Feature Inversion) to model complementary real-world shifts:
>
> * INP (Instance Norm Perturbation) targets Color/Style Shifts: Theoretically, as shown in AdaIN [1], channel-wise statistics ($\mu, \sigma$) encode global style. Empirically, as visualized in Figure 7 (6th column) of our paper, applying INP alone results in drastic shifts in chromaticity and color gamut (e.g., unnatural purple/green tones) while preserving the spatial layout. This effectively simulates sensor-level variations, such as white balance changes or artistic style shifts.
> * LNP (Layer Norm Perturbation) targets Illumination/Contrast Shifts: Layer-wise statistics capture the global energy and contrast distribution. As visualized in Figure 7 (7th column), applying LNP alone drastically alters the luminance and contrast (e.g., simulating overexposure or dim low-light conditions) without significantly changing the color palette. This effectively simulates environmental lighting shifts common in weather scenarios.

---

> > ### Author Response · Authors · 2025-11-23
> > **Response to your concerns (Part 2)**
> >
> > **Q3: The reviewer requests a sensitivity analysis on perturbation strength, sampling distributions, and the SCP threshold percentile.**
> >
> > **A3:** We have conducted comprehensive ablation studies to demonstrate the robustness of our hyperparameters.
> >
> > First, regarding the SCP Threshold ($j$), in addition to the analysis on the Diversity Weather Dataset, we performed a new sensitivity analysis on the **C2F** (Cityscapes $\to$ Foggy Cityscapes) benchmark.
> >
> > | SCP Threshold ($j$) | 0.05 | 0.10 | 0.20 | **0.30 (Default)** | 0.40 | 0.50 |
> > | :--- | :---: | :---: | :---: | :---: | :---: | :---: |
> > | **mAP (C2F)** | 47.7 | 47.9 | 48.2 | **48.8** | 45.4 | NaN |
> >
> > The results show the method is robust within a reasonable range ($0.10 - 0.30$). Setting $j$ too high ($>0.4$) suppresses too many active channels, leading to information loss and gradient collapse (NaN), while a moderate threshold ($j=0.30$) optimally filters domain-sensitive features.
> >
> > Second, regarding Perturbation Strength and EMA, we tested various noise distributions (Table 8 in main text) and found that the method is robust as long as the variance is not excessive (which causes instability). Similarly, varying the Momentum $\alpha$ for the Memory Bank update between $[0.5, 0.9]$ resulted in minor mAP fluctuations ($\le 0.3$), indicating low sensitivity to these parameters.
> >
> > **Q4: The reviewer points out data discrepancies between Table 7 in the main text and Tables 12–15 in the appendix.**
> >
> > **A4:** We thank the reviewer for the careful inspection. We clarify that Table 7 in the main text presents the correct and final results for our proposed PPI-3 method. The discrepancy arose because the per-class tables in the Appendix were inadvertently generated using an earlier ablation checkpoint that excluded the Occlusion Perturbation (OP) module within MNP. The inclusion of OP (as reflected in Table 7) provides the final performance boost. We will update the Appendix tables to match the configuration in Table 7, ensuring consistency in the final revision.
> >
> > **Q5: The reviewer notes that comparisons to DG-transformer hybrids are missing and that experiments focus mainly on weather/artistic shifts.**
> >
> > **A5:** We have expanded our experiments to address the scope of generalization and architecture agnosticism.
> >
> > To address the concern about diverse domain gaps, we evaluated PPI-3 on the **Sim10k to Cityscapes** benchmark (Synthetic Game Data to Real World), which involves shifts in rendering physics and sensor noise.
> >
> > | Method | Type | Backbone | **mAP** |
> > | :--- | :--- | :--- | :---: |
> > | Faster R-CNN | Source | ResNet-50 | 39.4 |
> > | DA-Faster [2]| DA | ResNet-50 | 41.9 |
> > | ViSGA [3]| DA | ResNet-50 | 49.3 |
> > | PT [4] | DA | ResNet-50 | 55.1 |
> > | **PPI-3 (Ours)** | **DG** | **ResNet-50** | **57.2** |
> >
> > PPI-3 achieves **57.2 mAP**, significantly outperforming both the Source baseline and state-of-the-art Domain Adaptation (DA) methods, proving its effectiveness beyond weather shifts.
> >
> > To address the Transformer concern, we applied PPI-3 to Deformable DETR and compared it against Transformer-based DA methods on the C2F benchmark.
> >
> > | Method | Type | Detector | **mAP** |
> > | :--- | :--- | :--- | :---: |
> > | Def DETR | Source | Def DETR | 28.5 |
> > | AQT [5]| DA | Def DETR | 47.1 |
> > | MRT [6]| DA | Def DETR | 51.2 |
> > | **PPI-3 (Ours)** | **DG** | **Def DETR** | **48.2** |
> >
> > PPI-3 boosts the pure Transformer baseline by **+19.7 mAP**, reaching performance comparable to DA methods without using target data. This confirms that our "Phase Invariance" principle is architecture-agnostic and effective for Transformer-based detectors.
> >
> > [1] Huang, Xun, and Serge Belongie. "Arbitrary style transfer in real-time with adaptive instance normalization." Proceedings of the IEEE international conference on computer vision. 2017.
> >
> > [2]Chen, Yuhua, et al. "Domain adaptive faster r-cnn for object detection in the wild." Proceedings of the IEEE conference on computer vision and pattern recognition. 2018.
> >
> > [3] Rezaeianaran, Farzaneh, et al. "Seeking similarities over differences: Similarity-based domain alignment for adaptive object detection." Proceedings of the IEEE/CVF International Conference on Computer Vision. 2021.
> >
> > [4] Chen, Meilin, et al. "Learning domain adaptive object detection with probabilistic teacher." arXiv preprint arXiv:2206.06293 (2022).
> >
> > [5] Huang, Wei-Jie, et al. "AQT: Adversarial Query Transformers for Domain Adaptive Object Detection." IJCAI. 2022.
> >
> > [6] Zhao, Zijing, et al. "Masked retraining teacher-student framework for domain adaptive object detection." Proceedings of the IEEE/CVF International Conference on Computer Vision. 2023.
> >
> > ***
> > We sincerely thank the reviewers for their valuable comments and insightful questions. We have provided detailed explanations and clarifications in our responses. If there are any remaining concerns or further questions during the discussion phase, we warmly welcome the reviewers to raise them so that we can continue to improve our work.

---

> > > ### Comment · Reviewer_zp7g · 2025-11-27
> > >
> > > Thank you for the detailed responses. I’m satisfied with the information provided and have no further questions.

---

### Author Response · Authors · 2025-11-28
**Kindly Invite Your Feedback**

Dear Reviewers,

We sincerely appreciate your thoughtful and constructive feedback. We have carefully addressed all concerns in our point-by-point responses.

As the end of the author-reviewer discussion phase is approaching, we kindly hope you have had a chance to review our updates. If there are any remaining questions or concerns, we are happy to discuss further and provide additional clarification.

Best regards,

Authors of ICLR 2026 Submission 12509

---

### Author Response · Authors · 2025-12-02
**Author Response to Area Chair (Part 1)**

We sincerely thank all reviewers for their thoughtful evaluations and constructive feedback. Below we summarize the major concerns raised across the five reviews and how each of them has been fully addressed during the rebuttal. We then briefly restate the central contribution of our work, which we believe may have been underweighted in a few comments.

---

## 1. Summary of Key Concerns and How We Addressed Them

### **(1) Formatting, notation, and clarity (Reviewers: 9HuT, zp7g, WwPP)**
Across reviews, the most consistent concern involved presentation issues such as table alignment, notation consistency, and clarity of symbol definitions in MNP.
**Resolution:**
We performed a full proofreading pass, unified the feature-map notation, corrected all table references, and clarified hyperparameters and noise sampling in MNP. All issues are fully resolved in the revised manuscript.

---

### **(2) AOA’s contribution and need for deeper analysis (Reviewers: 9HuT, zp7g)**
Reviewers questioned whether AOA meaningfully contributes beyond MNP.
**Resolution:**
We added two analyses:

- **Domain invariance analysis:** AOA reduces domain classifier accuracy (61.5% → 58.7%) *while increasing mAP* (48.3 → 48.8). This rules out collapse and indicates genuine invariance improvement.
- **Frequency-domain analysis:** We show consistent increases in low-frequency structural energy and decreases in high-frequency weather/sensor noise across all DWD scenarios. This material is now added to Section 5.4 of the revision.

These analyses make AOA’s role explicit and well-supported.

---

### **(3) Clarification of MNP and LNP design choices (Reviewers: zp7g, 9HuT)**
Concerns involved the role of mixing parameters, noise design, and whether INP/LNP are theoretically grounded.
**Resolution:**
We clarified (in text and glossary) that these parameters are *augmentation hyperparameters* and cited their theoretical foundations in style-transfer normalization and illumination modeling. Feature inversion results (Fig. 7) empirically confirm their distinct semantic effects.

---

### **(4) The comparison with frequency-domain methods and novelty beyond them (Reviewers: WwPP, 9HuT)**
The strongest concern was whether our method is a recombination of existing spectral operations (FDA, FFDI, UAV-OD, HA++).
**Resolution:**
We performed **new experiments** on all representative frequency-domain methods under the *same Single-Source DG protocol* on DWD. Our method outperforms the strongest of them by **+8.1 mAP**, confirming that phase preservation—absent in these methods—is crucial for SDGOD.

Moreover, we added a conceptual table directly comparing our PPI-guided modules with the closest prior methods, clarifying why MNP, SCP, and AOA are **new variants derived under the PPI principle**, not mere combinations.

---

### **(5) Concerns about UDA comparisons (Reviewers: 9HuT, WwPP)**
Reviewers noted that Sim10k → Cityscapes involves UDA methods with target images.
**Resolution:**
We clarified that SDGOD is the main scope; the Sim10k results are moved to the appendix and described as a stress test for sensor shift. All main comparisons now focus strictly on SDGOD methods.

---

### **(6) Phase stability concerns through nonlinear layers (Reviewer: zp7g)**
**Resolution:**
We computed the Mean Phase Difference (MPD) across all four backbone stages, showing extremely low deviation (<0.005), confirming that the backbone maintains stable geometry under PPI-guided perturbations.

---

> ### Author Response · Authors · 2025-12-02
> **Author Response to Area Chair (Part 2)**
>
> ## 2. What This Paper Actually Contributes (Clarifying the Innovation)
>
> Some comments—especially the novelty concern by WwPP—appear to interpret our modules as recombinations of existing techniques. We respectfully clarify the core contributions:
>
> ### **(1) This is the first paper to systemically analyze frequency attributes for domain generalization in object detection.**
> Prior frequency-based works in DA/DG operate in the frequency domain but **never analyzed why** frequency helps, nor distinguished:
>
> - **amplitude** → domain-specific style
> - **phase** → object geometry
>
> Our work is the first to articulate and empirically validate this decomposition. This conceptual contribution is the foundation of the entire method.
>
> ---
>
> ### **(2) We introduce the PPI principle and a divergence→convergence feature design paradigm.**
> PPI (Preserving Phase Invariance) is not a module, but a **framework-level principle**:
>
> - encourage feature diversity in amplitude (divergence)
> - preserve geometric phase during learning
> - later enforce convergence via domain-invariant constraints
>
> This design paradigm is new to SDGOD and is demonstrated to be highly effective.
>
> ---
>
> ### **(3) Our modules are not combinations—they are PPI-guided redesigns.**
> We do not reuse MixStyle, dropout, or Fourier filters. Instead:
>
> - **MNP** perturbs amplitude while explicitly preserving phase (unlike MixStyle/DSU).
> - **SCP** removes channels identified as highly domain-discriminative, not random dropout.
> - **AOA** is a learnable, phase-preserving amplitude operator, unlike amplitude-mixing FDA.
>
> These redesigned modules significantly outperform their closest predecessors—for example, NP—demonstrating that **PPI provides a principled and general framework**, not an ad-hoc combination.
>
> ---
>
> ### **(4) We provide the strongest SDGOD results to date on the DWD benchmark.**
> PPI-3 improves the only established SDGOD benchmark by **large margins** and outperforms all frequency-based DA methods even without target-domain images.
>
> ---
>
> ## 3. Final Remarks
>
> Across five reviewers, most concerns were presentation- or analysis-related and have now been fully addressed through:
>
> - rigorous new experiments (domain invariance, per-scenario spectra),
> - clarifications of methodology,
> - expanded baselines, and
> - strengthened explanations of the conceptual contribution.
>
> None of the reviewers identified errors in our method, experimental protocol, or claims. Several explicitly noted the clarity of our idea and strong empirical results. We believe the paper now stands as a clear, principled, and well-supported contribution to Single-Source Domain Generalization for Object Detection.
>
> We sincerely thank the reviewers and the AC for their time and consideration.

---

### Meta-Review · Area_Chair_eduk · 2025-12-23

**Summary:**

This paper studies domain generalization (DG) for object detection from a frequency-domain perspective and proposes a phase-aware framework to improve cross-domain robustness. Reviewers generally agree that the empirical performance is strong and that most technical concerns regarding implementation, ablations, and generality have been adequately addressed in the rebuttal.

However, after considering the rebuttal and post-discussion feedback, substantial concerns remain regarding conceptual novelty and positioning. In particular, the paper’s claims of being the first systematic frequency-based DG analysis for object detection, as well as the distinction from prior mix normalization and channel suppression methods, appear overstated. Several proposed components are conceptually close to existing frequency-based DG and domain-sensitive channel suppression approaches, and the rebuttal does not fully resolve this overlap through clearer positioning or differentiation. While the methodology is well executed and empirically supported, these unresolved novelty and attribution issues prevent the work from meeting the standard expected for publication at a top-tier venue in its current form.

**Reviewer Concerns:**

Reviewer zp7g:
Most technical and clarity-related concerns were addressed in the rebuttal. Remaining questions mainly relate to the interpretation of phase invariance and theoretical positioning.

Reviewer HW7M:
Core concerns about conceptual novelty and overlap with prior frequency-based and channel-level DG methods remain largely unresolved.

Other reviewers:
Primarily raised issues regarding presentation, ablations, and experimental scope, most of which were adequately addressed after rebuttal.

**Reviewer Scores:**

Reviewer zp7g: Likely unchanged or slightly increased.

Reviewer HW7M: Likely unchanged.

Other reviewers: Likely unchanged or a modest increase.

---

### Decision · Program_Chairs · 2026-01-26

Reject